# Conformational changes in the motor ATPase CpaF facilitate a rotary mechanism of Tad pilus assembly

Ian Y. Yen [1,2], Gregory B. Whitfield [3], John L. Rubinstein [1,2,4], Lori L. Burrows [5] ✉, Yves V. Brun [3] ✉ & P. Lynne Howell [1,2] ✉

The type IV pilus family uses PilT/VirB11-like ATPases to rapidly assemble and disassemble pilin subunits. Among these, the tight adherence (Tad) pilus performs both functions using a single bifunctional ATPase, CpaF. Here, we determine three conformationally distinct structures of CpaF hexamers with varying nucleotide occupancies by cryo-electron microscopy. Analysis of these structures suggest ATP binding and hydrolysis expand and rotate the hexamer pore clockwise while subsequent ADP release contracts the ATPase. Truncation of the intrinsically disordered region of CpaF in *Caulobacter crescentus* equally reduces pilus extension and retraction events observed using fluorescence microscopy, but does not reduce ATPase activity. AlphaFold3 modeling suggests that CpaF and other motors of the type IV filament superfamily employ conserved secondary structural features to engage their respective platform proteins. From these data, we propose that CpaF uses a clockwise, rotary mechanism of catalysis to assemble a right-handed, helical Tad pilus, a process broadly applicable to other single motor systems.

The type IV filament (TFF) superfamily is a group of nanomachines broadly distributed across bacteria and archaea[1,2]. These systems include the type IV pilus (T4P) family, the type II secretion system (T2SS), the Gram-positive competence (Com) pilus, and the archaellum. TFF machineries enable diverse functions including twitching and swimming motilities, natural competence, surface sensing and attachment, and protein secretion[3–7]. The T4P family is further subdivided into the type IVa pilus (T4aP), the type IVb pilus (T4bP), and the tight adherence (Tad) pilus[3,8]. A conserved and distinguishing hallmark of the T4P family is the dynamic assembly (extension) and disassembly (retraction) of protein subunits into helical filaments, or pili. However, the molecular details of pilus extension and retraction remain unknown.

T4P assembly and disassembly is catalyzed by the inner membrane motor subcomplex of the biosynthetic machinery. This subcomplex consists of cytoplasmic PilT/VirB11-like hexameric ATPases and inner membrane-embedded PilC-like platform proteins[9,10], both of which are conserved across the TFF superfamily[1]. In situ subtomogram averaging of the T4aP, T4bP, T2SS, and archaellar machineries demonstrated that the respective pairs of platform protein and ATPase interact within a socket composed of the alignment subcomplex[10–14]. This architecture suggests conformational changes in the motor ATPase due to ATP hydrolysis impart mechanical forces on the platform proteins, leading to processive pilin subunit extraction from or deposition into the membrane.

The prototypical system for studying filament assembly and disassembly across the TFF systems is the T4aP, which employs at least two monofunctional ATPases, PilB and PilT, dedicated exclusively to pilus extension and retraction, respectively. Multiple structures of PilB

[1]Program in Molecular Medicine, Peter Gilgan Center for Research and Learning, The Hospital for Sick Children, Toronto, ON, Canada. [2]Department of Biochemistry, University of Toronto, Toronto, ON, Canada. [3]Département de Microbiologie, Infectiologie et Immunologie, Université de Montréal, Montréal, QC, Canada. [4]Department of Medical Biophysics, University of Toronto, Toronto, ON, Canada. [5]Biochemistry and Biomedical Sciences and the Michael G. DeGroote Centre for Infectious Disease Research, McMaster University, Hamilton, ON, Canada. ✉e-mail: lori.burrows@mcmaster.ca; yves.brun@umontreal.ca; howell@sickkids.ca

homologues were determined solely with two-fold rotational ($C_2$) symmetry, whereas $C_2$, $C_3$, and $C_6$ hexamers have been reported for PilT[15–21]. These hexamers are characterized by the distinct nucleotide occupancies (ATP-bound, ADP-bound, or apo) of the six active site pockets, which indicates the direction of nucleotide turnover[15–21]. For instance, the $C_2$-symmetric structures of PilB and PilT from *Geobacter metallireducens* enabled the proposal of a rotary mechanism of catalysis, in which PilB conformational changes during ATP hydrolysis rotate PilC clockwise to polymerize the pilus, while PilT propagates the reverse rotation to depolymerize the filament[18,19]. However, most TFF systems encode a single motor ATPase that can assemble and disassemble the filaments in the absence of a dedicated retraction ATPase[1,5,6,22,23].

The Tad pilus in the freshwater bacterium *Caulobacter crescentus* is a single motor system[22]. Despite functional similarity to the T4aP, the Tad machinery is evolutionarily descended from archaeal T4P, which are also single motor systems[1]. The ATPase CpaF in *C. crescentus* is the only Tad motor reported to date that is known to be bifunctional, facilitating both pilus extension and retraction at the flagellar pole through ATP hydrolysis[6,22]. Tad ATPases, and by extension the Tad pilus locus, are widely distributed across bacterial phyla[1,8,22], and therefore constitute important enzymatic targets for mechanistic study. Indeed, structural information on the Tad machinery is beginning to emerge[24–27]; however, the fundamental question of how specific conformations of CpaF during catalysis could contribute to Tad pilus assembly and disassembly remains poorly understood.

As previous structural analyses of T4aP motor ATPases suggested a rotary mechanism of catalysis[18,19], we employed a similar structural approach with CpaF. During the course of this study, the cryo-electron microscopy (cryo-EM) structure of a *C. crescentus* CpaF hexamer was published in complex with a substrate analog, and used to propose a rotary mechanism of CpaF catalysis and Tad pilus assembly[25]. A nucleotide-dependent switching mechanism was suggested to mediate pilus disassembly, with the N-terminal intrinsically disordered region (IDR) allosterically modulating the direction of CpaF rotation[25].

Here, we build upon this foundation by presenting three additional cryo-EM structures of *C. crescentus* CpaF at various nucleotide concentrations, including the highest resolution cryo-EM structure of a PilT/VirB11-like ATPase determined to date. Analysis of monomeric chains as rigid packing units reveals an interchain contact that mediates nucleotide binding and enables the mapping of domain conformational changes between structures during catalysis. Fluorescent labelling of Tad pili in *C. crescentus* demonstrates that deletion of the IDR of CpaF has an equally negative effect on both pilus extension and retraction. AlphaFold3 modeling of the Tad and other TFF motor subcomplexes suggests that highly conserved ATPase secondary structural features are employed to engage their respective platform proteins. Together our data enable us to propose that conformational changes in CpaF facilitate a clockwise, rotary mechanism of catalysis to couple ATP hydrolysis to the mechanical force of Tad pilus synthesis, a process that may be broadly applicable across single motor TFF systems.

## Results

### Cryo-EM structures of *C. crescentus* CpaF

Tad pilus extension and retraction in *C. crescentus* is a rapid and highly dynamic process driven by the ATPase activity of CpaF[22]. This enzyme is predicted to adopt a quadripartite domain architecture, comprising a 79-residue IDR and three-helix bundle (3HB), followed by an N-terminal domain (NTD) connected to a canonical C-terminal ATPase domain (CAD) via a flexible linker (Fig. 1A). To visualize the catalytic steps, we recombinantly expressed and purified *C. crescentus* CpaF and prepared samples containing different nucleotide concentrations for single-particle cryo-EM structure determination. In the sample without exogenous nucleotides, we resolved two distinct $C_2$ symmetric

structures, which we termed "closed" and "compact" based on the distance between symmetrically positioned residue D116 on the 3HBs, to overall resolutions of 3.4 and 3.2 Å, respectively (Fig. 1B, Supplementary Figs. 1A, B, 2A, 3A, 4 and Table S1). Thirty-six percent of the hexameric particles contributed to the reconstruction of the closed state, while 64% adopted the compact state (Fig. 1C). Nucleotide densities were not observed in the cryo-EM maps.

Addition of an equimolar mixture of ATP and ADP at undersaturating conditions to purified CpaF resulted in two $C_2$ symmetric structures, one in the compact state and one in a new conformation that we term "expanded", at nominal resolutions of 2.8 and 3.2 Å, respectively (Fig. 1D, E, Supplementary Figs. 1C, 2B, 3B, 5 and Table S1). This compact structure is the highest resolution cryo-EM structure of a PilT/VirB11-like ATPase reported to date, enabling unambiguous assignment of ATP-$Mg^{2+}$ in the CAD of chains a and d, and ADP-$Mg^{2+}$ in chains b and e, from the sharpened map (Fig. 1E and Supplementary Fig. 1D). The expanded structure has an additional pair of ADP-$Mg^{2+}$ in chains c and f (Supplementary Fig. 1C, D). The particle ratio between the compact and expanded states is approximately 2:1 (Fig. 1C). Closed state particles were no longer observed upon the addition of nucleotides, suggesting this is strictly an apo conformation. The two independently obtained compact structures from the two datasets superimpose with an overall root mean square deviation (RMSD) of 0.8 Å (Supplementary Fig. 1E). Their structural similarity indicates that CpaF can adopt the compact conformation in the absence and presence of up to four nucleotides.

Saturating all nucleotide binding sites with excess ATP yielded another expanded structure at a global resolution of 3.3 Å (Fig. 1E, F, Supplementary Figs. 2C, 3C, 6 and Table S1). All particles contributed to the single reconstruction (Fig. 1C), and the nucleotide composition was identical to the expanded structure from the under-saturated dataset (Supplementary Fig. 1D). The presence of ADP, despite supplementing the sample with excess ATP, suggests purified CpaF harbors in vitro enzymatic activity. The expanded hexamers in the two different datasets superimpose with an RMSD of 1.5 Å (Supplementary Fig. 1F).

In all five consensus cryo-EM maps of the hexamer, densities corresponding to the 3HB, NTD, and CAD, but not the IDR, were visible. The 3HB densities were over-refined (emergence of artificial features in the map—see Methods section), suggesting flexibility that may be coupled with the preceding disordered region. This necessitated a local refinement strategy around the 3HBs and NTDs of each asymmetric trimer to improve map quality (Supplementary Figs. 3–7). When we measured the Cα distances between adjacent 3HBs in the trimer, we observed a stepwise increase between chains e and f from the closed to the compact state, as well as between chains f and a from the compact to expanded state (Supplementary Fig. 1B). This suggests that expansion of the 3HB, and thus the hexamer pore, is mediated by nucleotide occupancy. Since CpaF assumes three main conformations in a nucleotide-dependent manner, we believe the closed, compact, and expanded states captured in the apo, under-saturating, and saturating datasets, respectively, best represent the catalytic cycle. Our subsequent analyses described below focus on these three structures.

### The nucleotide identity of each chain defines the packing unit

To understand how the three CpaF structures contribute to the catalytic cycle, we grouped all eighteen chains according to their bound nucleotide and aligned the Cα atoms of all chains within each group. The ATP-bound, ADP-bound, and Apo monomers superimposed with an average RMSD of 1.2 Å, 1.2 Å, and 0.8 Å, respectively (Supplementary Fig. 8A). The overall intrachain conformation is conserved within, but not between groups, revealing nucleotide induced changes mediated by the hinge in the flexible linker (Supplementary Fig. 8B). For instance, while the 3HBs and NTDs of the ATP-bound and Apo chains align well, the CADs exhibit structural rotations. Interestingly,

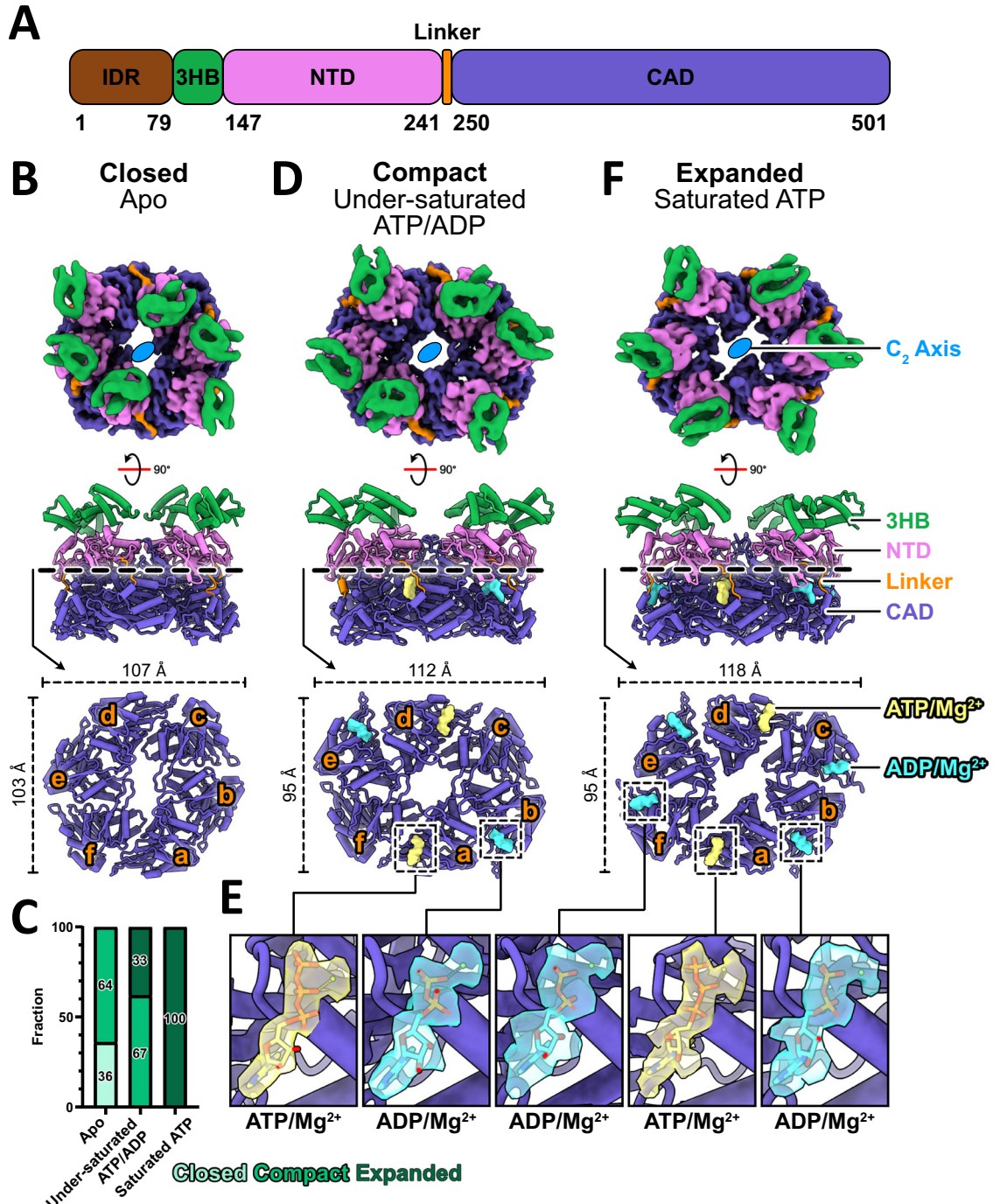

**Fig. 1 | Cryo-EM structures of *C. crescentus* CpaF. A** Domain architecture of CpaF. IDR intrinsically disordered region, 3HB three-helix bundle, NTD N-terminal domain, CAD C-terminal ATPase domain. **B**, **D**, **F** Structures of CpaF in the closed, compact, and expanded conformations determined from the Apo, under-saturated ATP/ADP, and saturated ATP datasets, respectively. (top) Top view composite maps were generated by combining the locally refined maps of the three-helix bundles with the NTD-CAD domains of the consensus maps. The IDRs were not resolved in any of the cryo-EM maps. (middle) Side view atomic models. (bottom) Top view cross-sections through the NTD and CAD to reveal the bound nucleotides in the CAD. Each monomeric CAD chain is labeled counterclockwise from a to f. **C** Distribution of particles that contributed to the reconstruction of each conformation for the three datasets. Increasing nucleotide concentrations favor the expanded state. **E** Close-up view of nucleotides in the active sites encased in their locally sharpened map densities. Densities are shown at the same thresholds within each dataset.

two chains in the closed state adopted an ATP-bound configuration despite the absence of ATP density in the unsharpened map. Lowering the threshold of the sharpened map, however, revealed residual density in the active site, which potentially fit a diphosphate (Supplementary Fig. 8A). It is possible that ATP was retained during purification, but that low nucleotide occupancy resulted in this fragmented density. Due to this ambiguity, coupled with the ATP-bound

configuration, we designated these two chains as ATP*. Overall, the three CpaF structures are composed of differing combinations of these four types of unique chains (Supplementary Fig. 8C).

CpaF oligomerization is mediated by interactions between the NTD of one chain and the CAD of an adjacent chain (Fig. 2A). Analysis of the active site pocket revealed that multiple residues from both chains are involved in nucleotide coordination. To better describe how

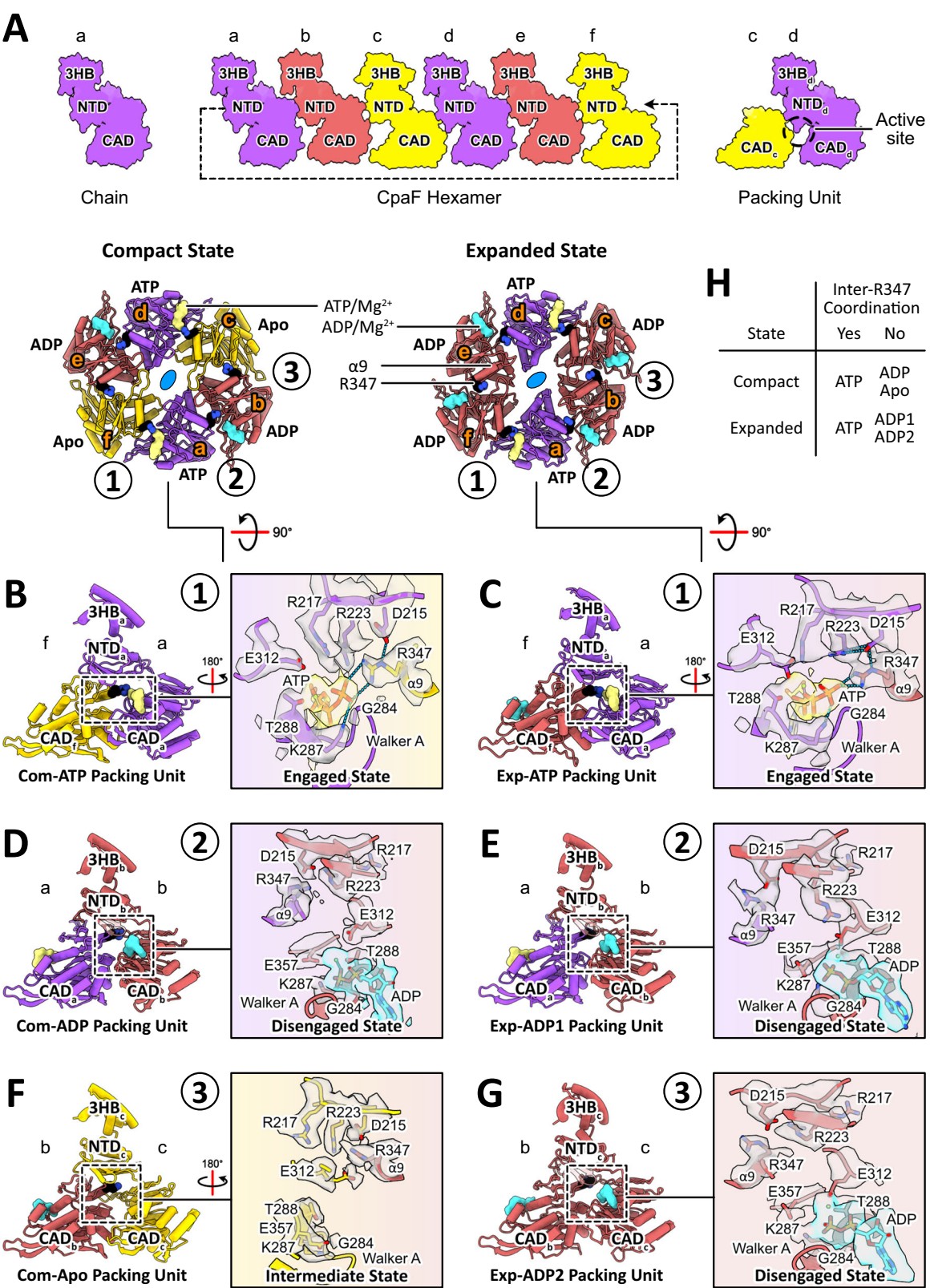

conformational changes within each pair of chains influence this interaction interface, we have employed the packing unit nomenclature, used previously to describe other PilT/VirB11-like ATPases[18,19,28,29], but have modified the definition to incorporate the entire chain of one monomer and the CAD of the adjacent monomer (Fig. 2A). Each packing unit is named according to the CpaF conformational state and the nucleotides bound between the CADs. For

instance, the ATP-bound packing unit of the compact state is termed Com-ATP. This approach enables us to explain how the relative domain orientations are impacted during nucleotide turnover.

## Interchain interactions mediate nucleotide binding

To understand how nucleotide binding is mediated in the three structures, we examined the active site pocket of each packing unit

**Fig. 2 | Interchain interactions mediate nucleotide binding. A** (left) A single polypeptide chain (hereafter referred to simply as a "chain") is comprised of the 3HB, NTD, and CAD. (middle) The CpaF hexamer consists of six chains mediated by interactions between the NTD of one chain and the CAD of an adjacent chain in a hand-to-shoulder configuration. (right) Packing units are defined as encompassing one full chain and the CAD of the adjacent chain. They are named according to their hexameric conformation and the bound nucleotide. Chains are labelled from a to f based on their positions within the hexamer. **B–G** Packing unit analyses for the compact and expanded structures. Packing units are numbered 1 to 3 according to their positions within an asymmetric unit of the hexamer. Full chains and their constituent domains are labelled from a to f and colored according to their bound nucleotides—ATP in purple, ADP in maroon, and Apo in gold. Atomic models of ATP in yellow and ADP in cyan are depicted in surface view while the residue R347 on the α9 helix in black is shown as a space-filling model colored by its side chain heteroatom. Active site analysis highlights residues, shown in stick representation, involved in stabilizing bound nucleotides. All residues and nucleotides are encased in their corresponding cryo-EM density (grey for amino acids, yellow for ATP, and cyan for ADP) contoured at the same threshold level within each dataset. Dotted blue lines denote hydrogen bonds. The side chain rotamers of R217, R223, and R347 are classified as "engaged", "disengaged", or "intermediate". **H** Summary of whether the bound nucleotide is coordinated by R347 of the adjacent CAD for all the packing units of the two structures.

within an asymmetric unit. In the ATP-bound packing units of the compact and expanded structures, Com-ATP and Exp-ATP, the α9 helix on the CAD of chain f is spatially aligned with and in proximity to the Walker A motif on the CAD of chain a, enabling hydrogen bond formation between an arginine finger R347 and the γ-phosphate of ATP, respectively (Fig. 2B, C). The acidic residue D215 on the NTD stabilizes R347, contributing indirectly to nucleotide binding. The canonical K287 of the Walker A motif coordinates the γ-phosphate, while the adjacent residue T288 and E312 of the Asp box coordinates and stabilizes the $Mg^{2+}$ ion, respectively. In addition, arginine fingers R217 and R223 on the NTD orient towards the active site to neutralize the negatively charged triphosphate. Thus, in the ATP-bound packing units, R347, R217, and R223 adopt a conformation that we term the "engaged state".

We next investigated the ADP-bound packing units of the compact and expanded structures—Com-ADP and Exp-ADP. Given the expanded state harbors two ADP-bound monomers in the asymmetric unit, we further classify Exp-ADP into Exp-ADP1 and Exp-ADP2. In the presence of ADP, the R347-mediated ligand interaction is absent and the α9 helix and Walker A motif in the CADs of chains a and b, respectively, are no longer in proximity to each other (Fig. 2D–E, G). The distance between these two secondary structural elements, measured as the Cα distance between the main chains of R347 and the Walker A motif G284, widens to 15.1 Å, compared to 8 Å in the ATP-bound packing units. Arginine fingers R347, R217, and R223 all orient away from the nucleotide binding pocket, now adopting what we term the "disengaged state". Despite the altered rotamer configuration, R347 is still stabilized by D215. K287 now interacts with the β-phosphate of ADP and E357 of the Walker B motif, while T288 and E312 remain coordinated to the $Mg^{2+}$ ion. Overall, this suggests that the hydrolysis of ATP facilitates residue level conformational changes between the ATP-bound and ADP-bound packing units.

Finally, in the Apo packing units of the compact and closed structures, Com-Apo, Clo-Apo1, and Clo-Apo2, designations similar to the Exp-ADP packing units above, the distance between the α9 helix and the Walker A motif in the CADs of chains b and c, respectively, increases to 17.4 Å, indicating that R347 moves progressively further away from the motif compared to the ADP-bound packing units (Fig. 2F and Supplementary Fig. 9B, C). Interestingly, the three arginine fingers adopt an "intermediate state" where R347 points away from the active site pocket towards the hexamer pore, while R217 and R223 project towards it, potentially primed to accommodate an incoming ATP. Given that E312 no longer coordinates a $Mg^{2+}$ ion, it is now positioned closer to the NTD than the Walker A motif. The arginine residues in the ATP* packing unit of the closed state, Clo-ATP*, assume a similar conformation, which suggests that, despite the ATP* monomer assuming an ATP-bound configuration, in the absence of a nucleotide, R347 undergoes rotamer changes (Supplementary Fig. 9A).

The packing unit analysis illustrates that interchain R347 coordination is observed only in ATP-bound packing units (Fig. 2H). Multiple sequence alignments of CpaF homologs across all major proteobacterial phyla, as well as of CpaF and other PilT/VirB11-like ATPases, revealed conservation of R347 and its equivalent in all TFF system

ATPases (Supplementary Fig. 10A, B). Overall, the packing unit analysis highlights how nucleotides are coordinated in the active site pocket, and how active site occupancy influences the conformation of CpaF chains.

## The CpaF hexamer expands and contracts during the catalytic cycle

Given that the compact and expanded structures harbor nucleotides, we believe these two conformations, but not the closed state, actively participate in catalysis. To dissect the catalytic cycle, the two structures were superimposed on the $C_2$ axis such that the corresponding monomers between the compact and expanded structures were aligned. This results in a clockwise axis rotation by 60°. Chains c and f adopt the Apo and ATP-bound conformations in the compact and expanded hexamers, respectively, which upon superimposition reflects an ATP-binding event (Fig. 3A). This also brought chains d and a, assuming the ATP- and ADP-bound conformations in the compact and expanded states, respectively, into alignment. This represents an ATP hydrolysis event (Fig. 3A). To capture an ADP release event, the compact structure was superimposed with the expanded structure on the same symmetry axis to bring chains e and b, adopting the ADP-bound and Apo conformations in the compact and expanded structures, respectively, into alignment (Fig. 3A). The conformational trajectories between aligned chains described above were analyzed as a packing unit. The domains within each pair of aligned packing units are now collectively referred to by their chain designation. For simplicity, only the three nucleotide events within the asymmetric unit are described, though they occur as a pair in the hexamer.

The cycle begins with an ATP molecule binding to a Com-Apo packing unit (chain f and its associated CAD from chain e), transitioning this packing unit into the Exp-ATP state (Fig. 3A, B). ATP binding induces an inward translation of the 3HB and NTD of chain f, $3HB_f$ and $NTD_f$, respectively, and the CAD of chain e, $CAD_e$, towards the hexamer pore and a dramatic 42° clockwise rotation of $CAD_f$ about the axis orthogonal to the hexamer plane, evident by the movement of the extended pore loop between the β9 strand and the α9 helix (Fig. 3B, Supplementary Fig. 11A and Supplementary Movie 1). Motions between the two CADs orient the Walker A motif of $CAD_f$ near the α9 helix of $CAD_e$, converting R347 from the disengaged to the engaged state to stabilize the γ-phosphate (Fig. 3C). The Cα distances of select residues: D116 of $3HB_f$, C196 of $NTD_f$, and G284 and E331 of the Walker A motif and extended pore loop in the CADs of chains e and f, respectively, were measured during the transition between the Com-Apo and Exp-ATP states (Fig. 3D). The 12 and 23 Å distance movement observed for G284 and E331 of $CAD_f$ are consistent with the dramatic conformational changes observed in this region.

Concurrent with ATP binding, the adjacent packing unit, Com-ATP (chain a and the CAD of chain f) hydrolyzes ATP to form the Exp-ADP1 packing unit (Fig. 3A). Packing unit alignment highlights how the 42° clockwise rotation of $CAD_f$ observed during ATP binding is coupled to a 30° clockwise rotation in $3HB_a$ and $NTD_a$ and a translation in $CAD_a$ away from the hexamer pore, with D116, C196, and G284 displaced by 24, 13, and 8 Å, respectively (Fig. 3D, E, Supplementary

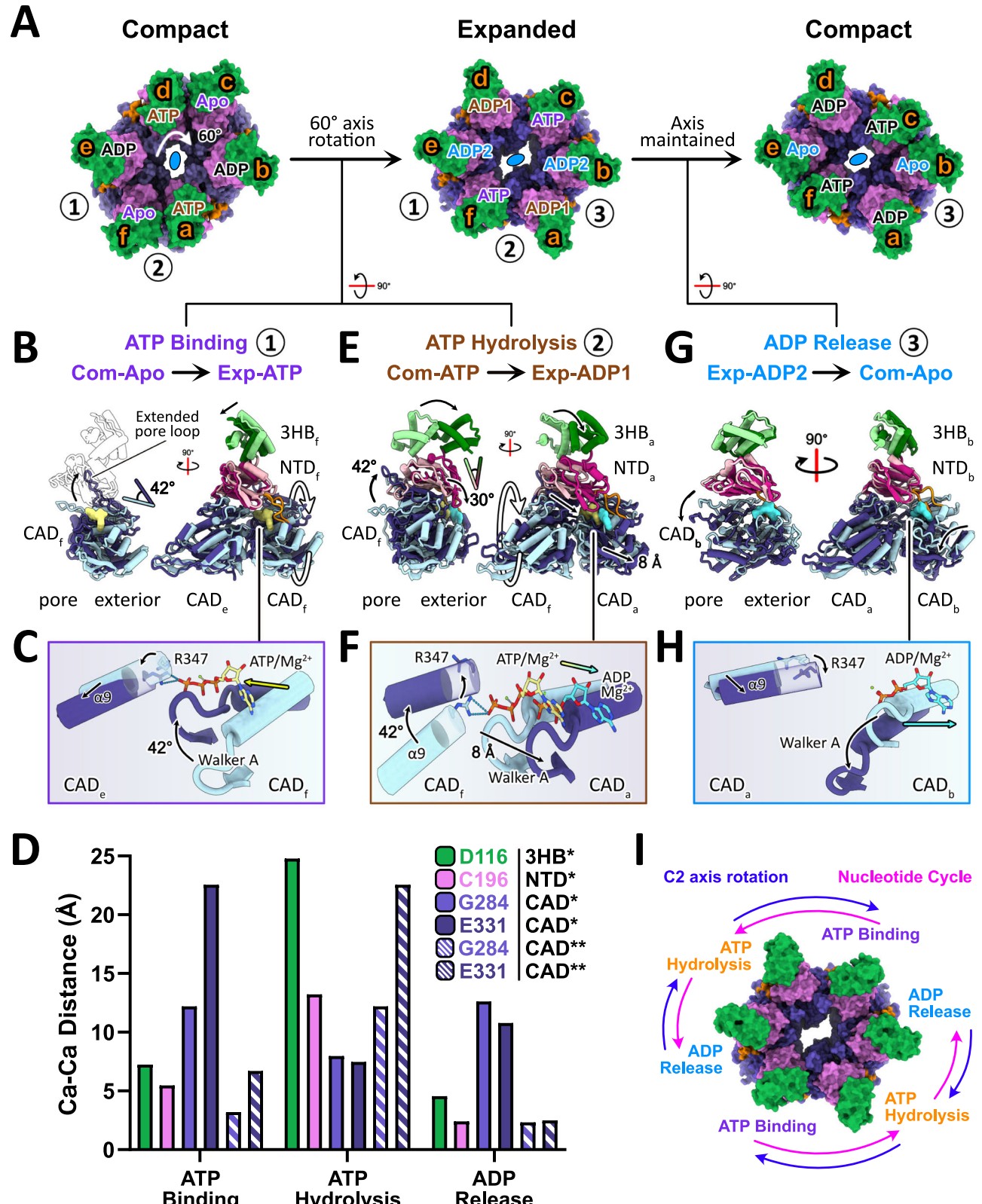

Fig. 11B and Supplementary Movie 2). Consequently, the coupled 42° rotation in $CAD_f$ elevates the α9 helix and reorients R347 into the disengaged state, while the 8 Å translation in $CAD_a$ shifts the Walker A motif (Fig. 3F). Together, this abrogates the R347-mediated coordination of the γ-phosphate of ATP, promoting hydrolysis. The $3HB_a$ and $NTD_a$ rotate outward as one rigid body, which expands the diameter of the hexamer pore, promoting the compact-to-expanded transition of

CpaF (Fig. 3A, E and Supplementary Fig. 11B). This correlates with the increased Cα distance between $3HB_f$ and $3HB_a$ and between the 3HB symmetric to $3HB_a$ (Supplementary Fig. 1B).

ADP release is required to complete the catalytic cycle. Given that ADP release is not coupled to nucleotide binding and hydrolysis in the compact-to-expanded transition, we reason that CpaF must undergo the reverse expanded-to-compact transition to release ADP (Fig. 3A).

**Fig. 3 | The CpaF hexamer expands and contracts during the catalytic cycle.**
**A** The compact-to-expanded-to-compact transition of the CpaF hexamer in one round of catalysis. Packing units are numbered from 1 to 3 according to their nucleotide event. Each chain is labelled from a to f and with its bound nucleotide, colored according to catalytic event: ATP binding in purple, ATP hydrolysis in brown, and ADP release in blue. For example, ATP and ADP1 in chain a of the compact and expanded states are both coloured brown to represent the hydrolysis of ATP to ADP. Nucleotides in black are not involved in catalysis during that cycle. Domains are coloured as depicted in Fig. 1A. The compact-to-expanded transition rotates the $C_2$ axis clockwise by 60° while the same axis is maintained when the hexamer contracts back to the compact state. **B**, **E**, **G** Superimposition of the packing units highlighting specific domain movements. The conformational trajectory goes from the light to the

dark packing unit. The domains within each pair of superimposed packing units are referred to by their chain position. In (**B** and **G**) (left side), $CAD_e$ and $CAD_a$ are omitted for clarity. **C**, **F**, **H** Motions of the α9 helix, R347 residue, Walker A motif, and nucleotides observed during the three catalytic events depicted through arrows. The conformational trajectory is light to dark blue. **D** Select Cα-Cα distances (Å) of residues associated with a specific domain to illustrate conformational changes those residues undertook during each catalytic event. The color scheme reflects that of Fig. 1A. *refers to residues in the full chain while **indicates residues in the adjacent CAD, colored with additional stripes. ATP binding involves chains e and f, etc. Exact measurements are reported in the Source Data file. **I** Model of nucleotide cycle versus rotational axis. Each nucleotide binding event turns the nucleotide cycle counterclockwise while rotating the two-fold axis clockwise.

Thus, the Exp-ADP2 and Com-Apo packing units (chain b and the CAD of chain a) were superimposed while maintaining the same axis. This reveals a downward motion in $CAD_b$, which lowers the position of the Walker A motif by 12 Å, thus allowing release of ADP (Fig. 3D, G, Supplementary Fig. 11C and Supplementary Movie 3). As $CAD_b$ moves to release ADP, the remaining packing units of the hexamer contract inward, reducing the diameter of the hexamer pore. The packing units that previously released ADP can now accommodate a new ATP molecule in subsequent rounds of nucleotide cycling.

In summary, the coupling of ATP binding and hydrolysis expands the hexamer and rotates the $C_2$ axis clockwise by 60°. Nucleotide release subsequently returns the hexamer to the compact state, maintaining the same axis and completing one round of catalysis (Fig. 3I and Supplementary Movie 4). From this analysis, we propose that CpaF operates via a clockwise, rotary mechanism of catalysis. Rotation of the $C_2$ axis is directionally opposite to that of the nucleotide cycle. These local conformational changes identified over the course of catalysis are likely relevant for assembling a right-handed Tad pilus[26] in the context of the motor subcomplex, given that a similar rotary mechanism was proposed for PilB[18].

## The cryo-EM structures represent the minimum CpaF domain architecture in bacteria

To understand the broad applicability of our cryo-EM structures and proposed mechanism of CpaF catalysis across bacteria, we sampled the architectural landscape of CpaF and constructed a phylogenetic tree using CpaF orthologs from all major bacterial phyla (Fig. 4A and Supplementary Data 1). CpaF orthologs can be categorized into three main classes based on protein architecture, as queried from the Alphafold Protein Structure Database[30]. Class I comprises orthologs without an N-terminal IDR. This class includes the gammaproteobacterium *Aggregatibacter actinomycetemcomitans* and represents the minimum domain architecture required across all bacterial phyla (Fig. 4A, B). *C. crescentus* CpaF is a member of class II, harboring an N-terminal IDR, while the class III architecture has an additional N-terminal forkhead-associated (FHA) domain that binds phospho-threonine peptides[31], connected to the IDR and the rest of the protein (Fig. 4A, B). *C. crescentus* and other alphaproteobacterial CpaF orthologs are almost exclusively found in class II. Outside of this clade, class I orthologs dominate while class III orthologs are mostly found in betaproteobacteria. Our phylogenetic and structural analyses revealed that while a subset of CpaF orthologs harbor an IDR, our cryo-EM structures captured the minimum and most common domain architecture of CpaF, suggesting that our proposed catalytic mechanism is broadly applicable.

## Truncation of the CpaF IDR negatively affects Tad pilus extension and retraction

As our phylogenetic analysis indicated that the majority of CpaF orthologs lack an IDR, we examined whether this region is necessary for CpaF activity and pilus synthesis. The IUPred3 server predicts that the first 79 residues of CpaF are intrinsically disordered[32]. Thus, we

generated three IDR truncation constructs, $CpaF^{66-501}$, $CpaF^{73-501}$, and $CpaF^{80-501}$ (Fig. 5A), purified them, and assayed for in vitro ATPase activity and thermal stability. Enzyme-coupled ATPase assays revealed that IDR truncation does not abrogate enzyme activity while a Walker A motif point mutant $CpaF^{K287A}$ does (Fig. 5B). As more residues of the IDR are removed, there is a trend towards increased ATPase activity (Fig. 5B). A thermal shift assay demonstrated a difference of less than 1 °C in protein stability between the full length and all mutant constructs (Fig. 5C). Thus, the absence of the IDR does not reduce intrinsic enzymatic activity or thermal stability of CpaF.

To investigate whether the IDR impacts pilus biogenesis, we made equivalent truncations in *C. crescentus* at the native chromosomal *cpaF* locus. Extended pili of cells harboring a cysteine knock-in mutation in the major pilin PilA (*pil*-cys) were fluorescently labelled[6,33]. Upon pilus retraction, internalization of labelled pilins results in cell body fluorescence, indicating that a pilus extension and retraction event has occurred, which can be quantified by fluorescence microscopy, as shown previously[6,33]. Approximately 25% of parental (*pil*-cys) cells in a mixed population exhibited cell body fluorescence (Fig. 5D). This decreased significantly, to approximately 12%, when the IDR was completely removed, and decreased further if the IDR was incompletely truncated, down to 0.8% (Fig. 5D). Addition of PEG5000-maleimide during labeling physically blocks pilus retraction, producing cells with fluorescent pili and unlabeled cell bodies, which enables quantification of pilus extension events[6,33]. When we blocked retraction with PEG5000-maleimide and quantified cells with extended, labeled pili, a trend similar to that of the fluorescent cell body quantification experiments was observed (Fig. 5E). To determine whether the reduction in pilus activity was related to changes in CpaF cellular levels due to IDR truncation, cell lysates were probed by Western blot using CpaF-specific antibodies, which revealed that all three truncation mutants had reduced whole cell levels compared to the parental strain, and that these levels correlated with the observed pilus activity phenotypes (Supplementary Fig. 12A). Interestingly, further truncation of CpaF to remove both the IDR and 3HB, $CpaF^{145-501}$, $CpaF^{147-501}$, and $CpaF^{149-501}$, abolished expression of the protein (Supplementary Fig. 12A).

Since it was unclear whether the reduced pilus activity phenotypes of the IDR truncation mutants were due specifically to a functional role of the IDR, or to reduced CpaF levels in the cell, we overexpressed the truncation mutants from an inducible plasmid in a *cpaF* knock-out background. Western blot analysis of whole cell lysates revealed comparable protein expression levels among the truncation mutants, which were elevated above the native level of CpaF expression (Supplementary Fig. 12B). Under these conditions, all three truncation mutants exhibited similar pilus activities, either through quantification of fluorescent cell bodies (Fig. 5F and Supplementary Fig. 12B) or blocked pili (Fig. 5G and Supplementary Fig. 12C), suggesting that the IDR length-dependent changes in pilus activity observed previously were a consequence of the difference in protein levels between the truncation mutants. However, pilus activity of all the truncation mutants was significantly lower than when the full-

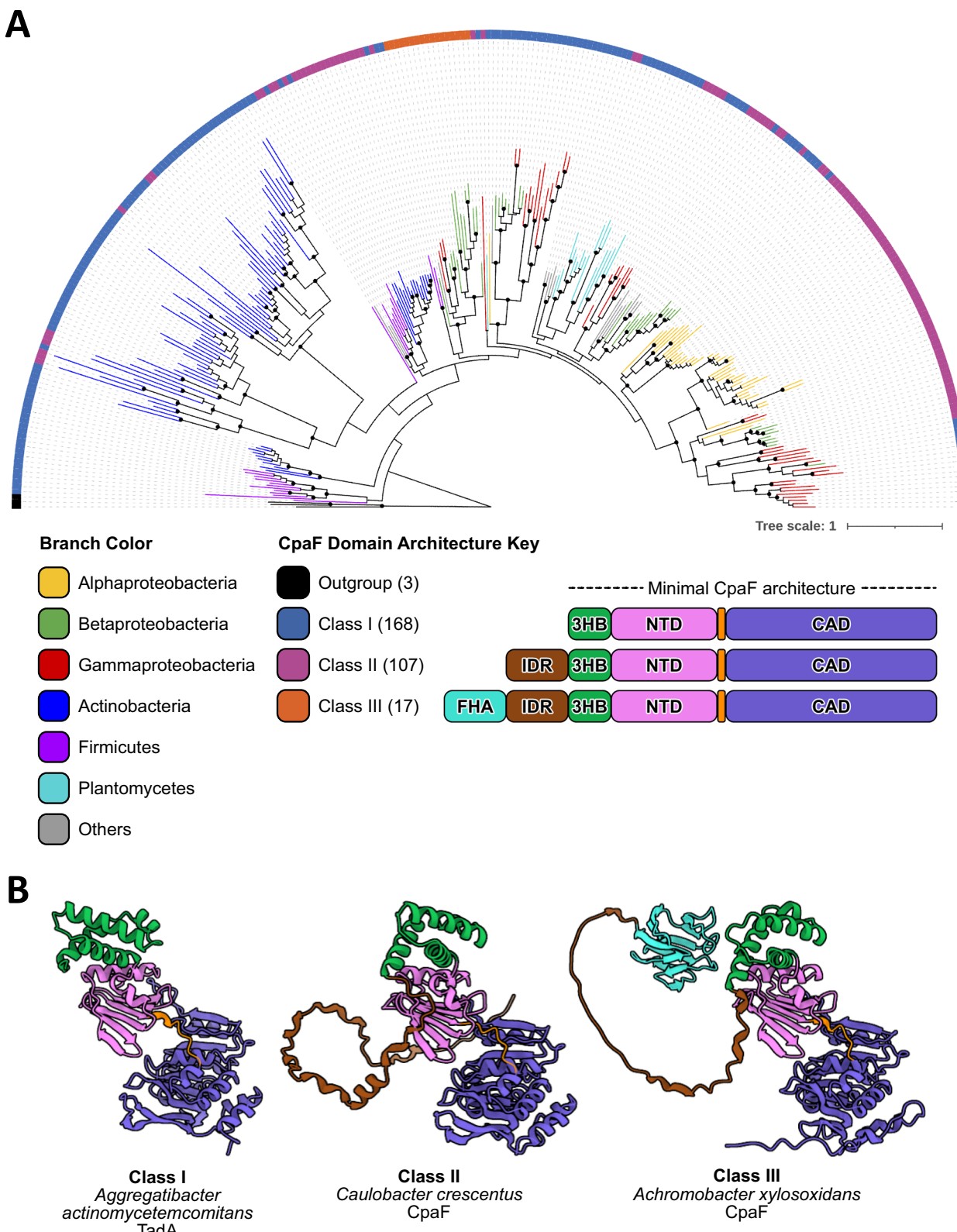

length protein was expressed at a similar level, indicating a potential role for the IDR in maximizing pilus dynamics (Fig. 5F, G). Finally, given that blocked pili correspond to pilus extension events during the period of pilus blocking and labelling, and fluorescent cell bodies are generated from pilus retraction events over the same labeling period, our data collectively suggest that IDR truncation affects Tad pilus extension and retraction equally.

## AlphaFold3 prediction of the Tad motor subcomplex

To understand how the proposed catalytic mechanism of CpaF is coupled to mechanical forces that power pilin polymerization and depolymerization, we leveraged AlphaFold3 to predict the structure of the Tad motor subcomplex[34]. The Tad operon encodes two PilC-like platform proteins, CpaG and CpaH, each containing a conserved, cytoplasmic type II secretion system F domain (T2SSF) that comprises

**Fig. 4 | CpaF phylogeny coupled to structural prediction reveals three CpaF domain architectural classes. A** Phylogenetic tree of 292 CpaF orthologs and 3 archaellar ATPases generated by RAxML maximum-likelihood analysis. Branches are colored according to bacterial phyla, while nodes with rapid bootstrap values greater than or equal to 80 are indicated by black circles. Black branches depict the archaellar ATPase outgroup. The tip of each leaf is associated with one of three classes of CpaF domain architecture, based on the predicted structure of each sequence from the AlphaFold Protein Structure Database. The three CpaF domain architecture classes are depicted in the legend along with the number of orthologs in each class. Accession numbers of each CpaF ortholog can be found in Supplementary Data 1 and the tree can also be viewed at the following link: https://itol. embl.de/tree/1322042512522045617219208 06. FHA forkhead-associated domain, IDR intrinsically disordered region, 3HB three-helix bundle, NTD N-terminal domain, CAD C-terminal ATPase domain. **B** Representative CpaF orthologs from each class and their AlphaFold predicted structures retrieved from UniProt, colored according to the domain architecture in (**A**). UniProt accession numbers: E1CIZ1 (*Aggregatibacter actinomycetemcomitans* TadA), Q9L714 (*Caulobacter crescentus* CpaF), and E3HIW0 (*Achromobacter xylosoxidans* CpaF).

a six-helix bundle[35]. Given that the platform protein of all other TFF systems has two T2SSF domains, we hypothesized that a CpaG-CpaH heterodimer functions analogously to a single PilC-like platform protein. Recent studies of the T2SS and Com pilus platform proteins PulF and PilC, respectively, suggest that these proteins form a homotrimeric complex, and we propose that there is a similar arrangement of CpaGH heterotrimers in the Tad system[36,37]. Our prediction comprised a hexamer of CpaF without the IDRs, bound to two ATPs, four ADPs, and six Mg$^{2+}$ ions to mimic the expanded state, and a heterotrimer of CpaG and CpaH with a seven-subunit PilA filament. The AlphaFold prediction returned a C$_6$ symmetric structure of CpaF with all chains adopting the same conformation despite differences in bound nucleotides, a C$_3$ symmetric heterotrimer of CpaGH, and a non-helical PilA filament (Supplementary Fig. 13A, B). Thus, we manually replaced the AlphaFold3 predicted CpaF and PilA structures with their respective cryo-EM structures of the expanded state and PilA filament from *C. crescentus*[26], to generate a composite experimental/predicted model of the Tad motor subcomplex (Fig. 6A).

To investigate how the platform proteins are predicted to interact with the ATPase, we examined the binding interface using the AlphaFold3 predicted model. Each CpaG or CpaH subunit directly interacts with one CpaF monomer, specifically between the T2SSF domains of CpaG or CpaH and the α5 helix (N189-V200) of CpaF and its downstream loop 8 (S201-P212), prior to the β4 strand of the NTD (Fig. 6B). The tip of the CpaF extended pore loop (T325-T337) also contacts the T2SSF domains (Fig. 6B). CpaF sequence alignments and residue conservation analyses revealed several conserved residues in these regions, including G204, R205, R206, and D208 on loop 8, and E331 and G334 on the extended pore loop (Supplementary Fig. 10C). In our cryo-EM structures, the α5 helix and loop 8 are positioned at similar heights in both the compact and expanded hexamers (Fig. 6C). However, the three extended pore loops within the asymmetric unit of both structures exhibit variable differences in height, ranging from 17 to 26 Å, with progressively greater distances toward the loop of chain a, which harbors an ATP (Fig. 6C). Only the extended pore loops in the ATP packing units can engage the T2SSF domains of CpaG and CpaH, suggesting that ATP binding events promote loop engagement, perhaps to stop CpaGH rotation. Given that these secondary structures are located in the ATPase pore, the ~80 Å diameter T2SSF heterotrimer must fit into the pore to engage these elements. The pore of the expanded and compact states can accommodate the platform as the 3HBs expand, though some conformational rearrangements of the platform heterotrimer may be required in the compact state.

The AlphaFold3 predicted Tad motor subcomplex also suggested that the 3HBs of CpaF interact with less conserved, exterior N-terminal helices and loops of CpaG and CpaH, though the confidence scores for these interactions are lower. Coloring the compact and expanded cryo-EM structures by electrostatic Coulombic potential revealed a highly negative patch on the 3HB, predicted to interact with a positively charged patch formed by α3 and α4 of CpaG and α4 of CpaH (Fig. 6D). The outward rotation of the 3HBs during catalysis (Fig. 3D, E and Supplementary Fig. 11B) could influence the conformations of these N-terminal helices and ultimately the platform complex.

The C-terminal α-helices of CpaG and CpaH are predicted to form a C$_3$ symmetric, concave shaft which likely accommodates the PilA

filament. Our composite model indicates that the curvature of the platform protein α-helices parallels the curved N-terminal helix of PilA, allowing one CpaGH pair to interact with one major pilin subunit (Fig. 6E). Given the predicted trimeric platform assembly and the helical organization of the PilA filament, it is possible that three pilins can be simultaneously assembled or disassembled. Each CpaGH heterodimer is separated by an approximately 22 Å wide opening that is predicted to be confluent with the inner membrane, which could facilitate entry into or exit from the core of the platform complex of a roughly 10 Å wide curved pilin (Fig. 6E). The N-terminal α1 of CpaH in the AlphaFold prediction is positioned in front of this opening (Fig. 6E). Given that the N-terminal helices of CpaG and CpaH are predicted to interact with the 3HB of CpaF, 3HB flexibility impacted by catalysis could move these helices, including α1 of CpaH to gate this opening (Fig. 6E). The motor subcomplex prediction implies that the CpaF hexamer and the CpaGH heterotrimer share multiple binding interfaces that may couple enzyme-induced conformational changes in CpaF to the platform proteins, which in turn facilitate interactions with major pilins and/or the pilus filament.

## Structural predictions of other motor subcomplexes in the TFF superfamily

To probe whether the ATPase-platform interface from the Tad motor subcomplex is conserved across other TFF systems, we predicted structures of the motor subcomplexes for the extension- and retraction-specific states of the T4aP, the T4bP, the archaellum, and the T2SS (Fig. 7A, Supplementary Fig. 13 and Supplementary Table 2). For each system, a hexamer of the ATPase and a homotrimer of the platform protein was supplied for the prediction. The core ATPase domains exhibit higher pLDDT confidence scores relative to the platform proteins, presumably because there are more PilT/VirB11-like ATPases and structurally similar AAA$^+$ ATPase structures deposited in the Protein Data Bank (Supplementary Fig. 13A). Apart from the retraction-specific PilT-PilC complex, the same secondary structural elements of each respective TFF ATPase, equivalent to the α5 helix, loop 8, and the extended pore loop of the Tad ATPase, engage their respective platform proteins (Fig. 7B). In contrast, PilT is predicted to contact PilC via its conserved AIRNLIRE motif on the CAD[38] (Fig. 7B). Overall, these predictions reveal that extension ATPases from the TFF superfamily may employ a similar platform protein binding interface, potentially unifying the TFF motor architecture.

## Discussion

Herein, we present three distinct cryo-EM structures of the *C. crescentus* Tad pilus ATPase CpaF in different nucleotide-bound states. Chain organization into rigid packing units reveals nucleotide coordination by interchain residues and facilitates mapping of domain movements during the sequential turnover of nucleotide. These structural interpretations enable us to propose a rotary mechanism of catalysis. Phylogenetic analysis of CpaF orthologs indicates that our cryo-EM structures represent the minimum and most common CpaF domain architecture, suggesting the broad applicability of our proposed mechanism. Truncations of the IDR do not reduce CpaF ATPase activity nor protein stability in vitro, but do negatively impact pilus biogenesis in *C. crescentus*, suggesting a more complex role for this

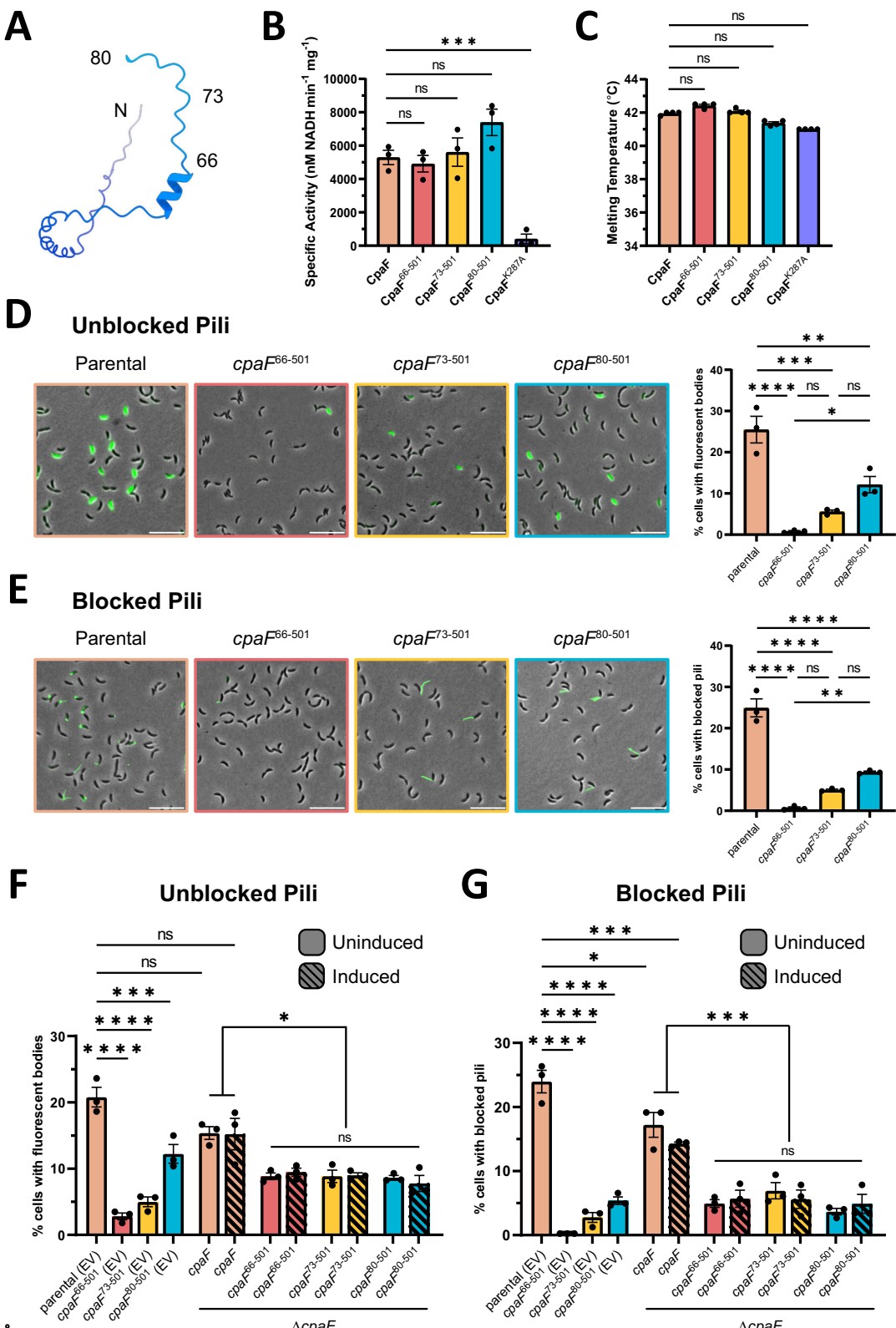

domain in pilus biology. Finally, our model of the Tad ATPase-platform interface allows us to identify a conserved region in CpaF that contacts the platforms. We extend this analysis to other TFF motor sub-complexes and found that extension ATPases use a similar binding site to engage their respective platform proteins. Our collective results provide a foundation for dissecting the mechanistic details of pilus dynamics in single motor systems across the TFF superfamily.

Our experimental approach in which CpaF was vitrified either without nucleotide or with nucleotide concentrations that would under- or fully saturate all nucleotide binding sites, yielded three distinct, high-resolution structures, which we termed "closed", "compact", and "expanded" (Fig. 1B–F). During preparation of this manuscript, a similar structural study of CpaF was published, where the protein samples were saturated with either the non-hydrolyzable

**Fig. 5 | Removal of the CpaF IDR negatively impacts pilus extension and retraction in vivo. A** AlphaFold predicted structure of the *C. crescentus* CpaF IDR, retrieved from UniProt under accession number Q9L714, depicting the three IDR truncation constructs' start positions. **B** In vitro enzyme-coupled ATPase assay. Mean ATPase activity was depicted as the change in nM of NADH per min per mg of protein. **C** In vitro thermal shift assay. Melting temperature was determined from four biological replicates. For (**B**, **C**), each data point represents one biological replicate with three technical replicates. Bar graphs indicate the mean ± SEM. Statistics were determined using One-Way ANOVA followed by Dunnett's multiple comparisons test comparing CpaF to mutants. ***$p = 0.0002$. ns not significant. **D** (left) Representative images of a mixed population of *C. crescentus pil-cys* (parental) cells harboring the indicated IDR truncations of *cpaF* at the native chromosomal locus labeled with AF488-maleimide. (right) Quantification of cell body fluorescence. **E** (left) Representative images of a mixed population of *C. crescentus*

*pil-cys* (parental) cells harboring the indicated IDR truncations of *cpaF* at the native locus, blocked with PEG5000-maleimide and labeled with AF488-maleimide. (right) Quantification of cells with blocked pili. Scale bars, 10 µm. **F** Quantification of cell body fluorescence in a mixed population of *C. crescentus pil-cys* cells (parental) labeled with AF488-maleimide dye. **G** Quantification of extended pili in a mixed population of *C. crescentus pil-cys* (parental) cells blocked with PEG5000-maleimide and labeled with AF488-maleimide dye. In (**F**, **G**), strains with *cpaF* at the native chromosomal locus were transformed with empty vector (EV), while strains in a Δ*cpaF* background were transformed with a taurine-inducible vector harboring full-length *cpaF* or the indicated truncation variants. For (**D**–**G**) the data represent three biological replicates, each data point represents one biological replicate with three technical replicates. Bar graphs indicate the mean ± SEM. Statistics were determined using One-Way ANOVA followed by Tukey's multiple comparisons test. Measurements and *P* values are reported in the Source Data file.

ATP analog AMP-PNP or ADP[25]. Consequently, only structures equivalent to our expanded state conformation, which they termed CpaF$_{AMP-PNP}$ and CpaF$_{ADP}$, were resolved. Notably, the positions of the bound ADPs and AMP-PNPs are inconsistent between our expanded and their CpaF$_{AMP-PNP}$ structures (Supplementary Fig. 14). This discrepancy may be explained by the four HEPES molecules trapped within four packing units in the CpaF$_{AMP-PNP}$ structure, which could potentially influence local nucleotide binding[25]. We did not observe density for HEPES molecules in our maps. The crystal structure of *G. metallireducens* PilT similarly captured ethylene glycol molecules within the packing unit, a site that has been proposed as a potential drug target to perturb catalysis[19]. This suggests small molecules could impact nucleotide binding. Our experimental approach offers three alternative CpaF structures at higher resolutions, with the compact and expanded states capturing nucleotides in previously unobserved positions.

Organization of individual chains into rigid packing units revealed interchain coordination of nucleotides. Specifically, arginine residues R217 and R223, as well as R347 from an adjacent monomer in the packing unit, adopt "engaged", "disengaged", and "intermediate" states in ATP-bound, ADP-bound, and Apo active sites, respectively, by altering their rotamer conformation (Fig. 2). Although interchain coordination of nucleotides is common among AAA$^+$ ATPases[39], we provide the structural observation of direct interchain coordination of ATP in PilT/VirB11-like ATPases. Interestingly, within the AMP-PNP packing units of the CpaF$_{AMP-PNP}$ structure, all three arginine residues adopt a conformation equivalent to our disengaged state[25]. We speculate that this is the consequence of AMP-PNP occupying the equivalent of an ADP packing unit. Indeed, the main chain Cα distance between R347 and G284 is 15.9 Å in the AMP-PNP packing unit of CpaF$_{AMP-PNP}$, consistent with the 15.1 Å distance in our ADP packing units. Even when both structures adopt similar hexameric and packing unit conformations, we observe distinct nucleotide binding arrangements in CpaF.

Interchain nucleotide coordination has been documented in the crystal structures of *Aquifex aeolicus* PilT and the archaellar ATPase FlaI, in which the R347-equivalent residues R207 and R326 stabilize a hydrolyzed γ-phosphate, respectively[20,29]. Both arginine residues orient away from the active site pocket when coordinated to the phosphate molecule. Given that R347 of CpaF in the disengaged state adopts a similar rotamer as FlaI and PilT, it could potentially coordinate hydrolyzed γ-phosphates as well. This may also suggest that its side chain follows the trajectory of the released γ-phosphate as it exits the hexamer. In addition, point mutations of the R207-equivalent residue in *Pseudomonas aeruginosa* PilT abrogated twitching motility, correlating the role of arginine fingers with pilus dynamics[20]. Given that this arginine residue is conserved across PilT/VirB11-like ATPases (Supplementary Fig. 10B), interchain nucleotide coordination is likely a common mechanism.

The shared evolutionary lineage of the Tad pilus and archaellum manifests as structurally similar domain architectures between CpaF

and FlaI, including the 3HB, NTD, and CAD. Our phylogenetic study of CpaF indicates that these three domains constitute the minimum domain architecture (Fig. 4). Indeed, deletion of the 3HB in both CpaF and FlaI resulted in null protein expression in *C. crescentus* (Supplementary Fig. 12A) and abrogation of archaellar assembly[29], respectively. However, structural variability at the N-terminus, for example the IDR in *C. crescentus* CpaF or the 29-residue folded domain in FlaI, may correspond to specific functional adaptations. The N-terminal 29 residues of FlaI are essential for swimming motility, but not archaellar assembly[29]. Given that the Tad pilus does not facilitate swimming in *C. crescentus*, the N-terminal folded domain of FlaI was lost, as observed in class I CpaF orthologs, or was otherwise replaced with a different domain that provides a more relevant functionality, as for class II and III orthologs. This explains why truncating the IDR of *C. crescentus* CpaF did not abrogate in vitro ATPase activity (Fig. 5B), which only involves the core NTD-CAD, but did negatively affect pilus extension and retraction (Fig. 5D–G). How this domain participates in this process remains to be determined.

The IDR of CpaF was previously proposed to modulate the bifunctionality of CpaF by allosterically gating the active site pockets and thus, influencing nucleotide binding[25]. Phylogenetic analysis reveals that the majority of CpaF orthologs lack an IDR (Fig. 4), including the Tad ATPase of the commensal bacterium *Micrococcus luteus*. Tad pili in *M. luteus* are implicated in natural transformation, a process that requires pilus extension and retraction[40]. This suggests that the IDR is likely not the common element that dictates bifunctionality across CpaF orthologs. Furthermore, quantification of fluorescent cell bodies and importantly, blocked pili, reveals that *C. crescentus* can still extend and retract pili when the IDR of CpaF is removed, albeit less frequently compared to the parental strain (Fig. 5D–G). These examples are inconsistent with the IDR modulating CpaF bifunctionality. When *C. crescentus cpaF* was cross-complemented in the alphaproteobacterium *Asticcacaulis biprosthecum* and vice versa, variations in pilus activity were observed, which was attributed to differences between the two IDR protein sequences[22]. We speculate that the IDR could mediate protein-protein interactions with components of the biosynthetic machinery to regulate pilus activity.

Combining our CpaF catalytic mechanism (Fig. 3) and composite model of the Tad motor subcomplex (Fig. 6A), we propose that the coupling of ATP binding and hydrolysis expands the CpaF hexamer and simultaneously rotates the platform complex clockwise by 60° (Figs. 3 and 8). Given that only the extended pore loop of the ATP-bound chain is sufficiently elevated to engage the platform proteins, we hypothesize that the upward thrust in the CAD of CpaF during ATP binding could mediate interactions with CpaG and CpaH to stop their rotation (Figs. 3B and 6B, C). The predicted C$_3$-symmetric CpaGH heterotrimer fits snugly into the hexamer pore of the CpaF expanded state, suggesting that in this ATPase conformation, the platform complex could facilitate entry of three pilin subunits from the inner

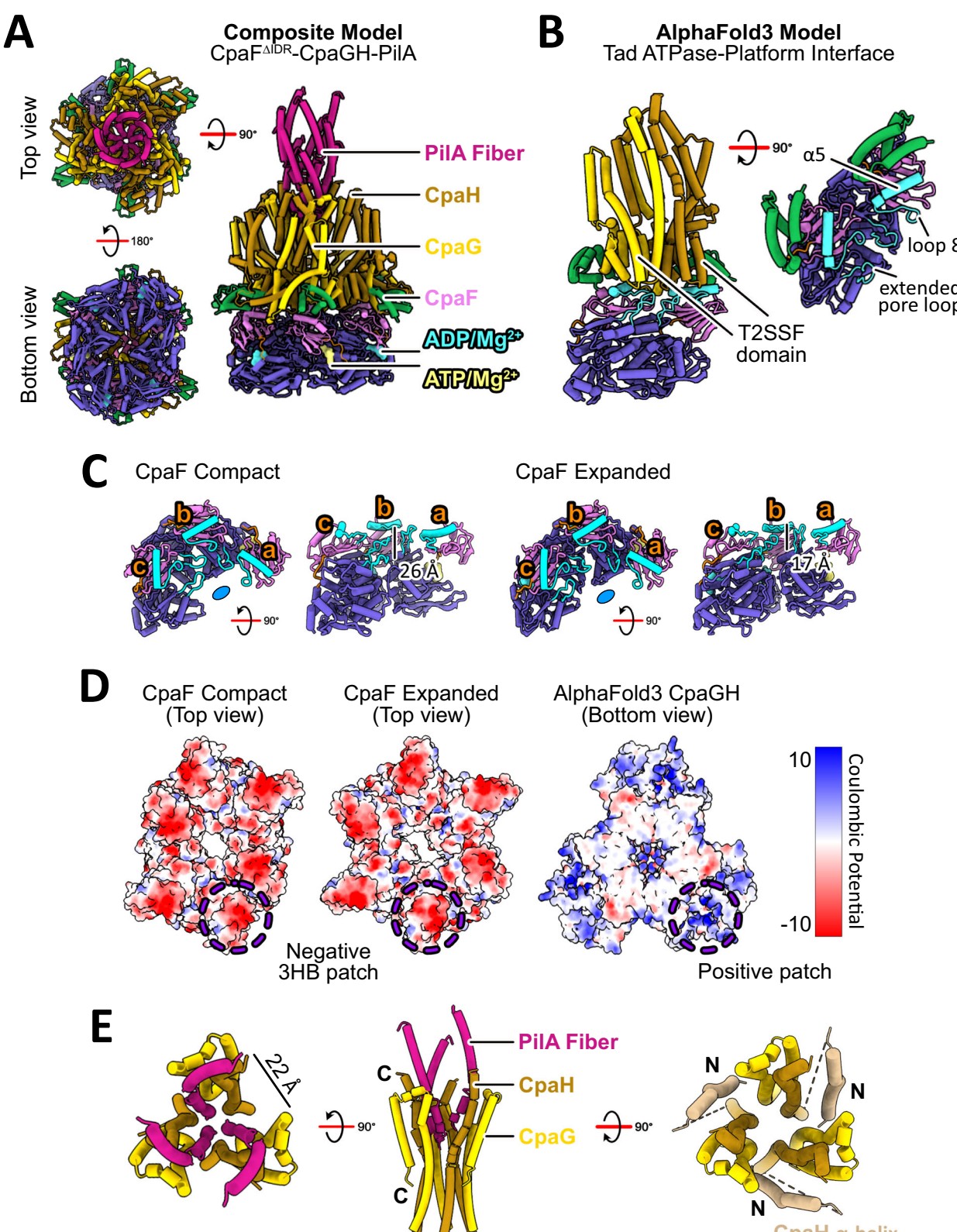

**Fig. 6 | AlphaFold3 prediction of the Tad motor subcomplex. A** Composite model of the Tad motor subcomplex in which AlphaFold predicted CpaF and PilA were replaced and manually docked with the cryo-EM expanded structure of CpaF and the PilA filament (PDB: 8U1K), respectively. **B** AlphaFold3 predicted Tad ATPase-platform interface depicting secondary structural elements α5 (N189-V200), loop 8 (S201-P212), and the extended pore loop (T325-T337) in CpaF that engage the T2SSF domains of CpaG and CpaH. **C** Height differences in the extended pore loops within the asymmetric unit of each structure. Each chain is labelled from a to c. The height

between the extended pore loop of chain a and c is indicated for each structure. **D** Expanded and compact CpaF structures and the AlphaFold predicted platform complex colored by Coulombic potential, contoured from +10 (blue) to −10 (red) kT/e. **E** Simplified view of the composite model in which the AlphaFold predicted heterotrimer CpaG-CpaH forms a central shaft to accommodate three PilA subunits from the PilA filament structure. Each CpaGH heterodimer is separated by a ~ 22 Å gap that is mediated by the N-terminal helix of CpaH to facilitate PilA entry into or exit from the shaft within the inner membrane. C C-terminal end, N N-terminal end.

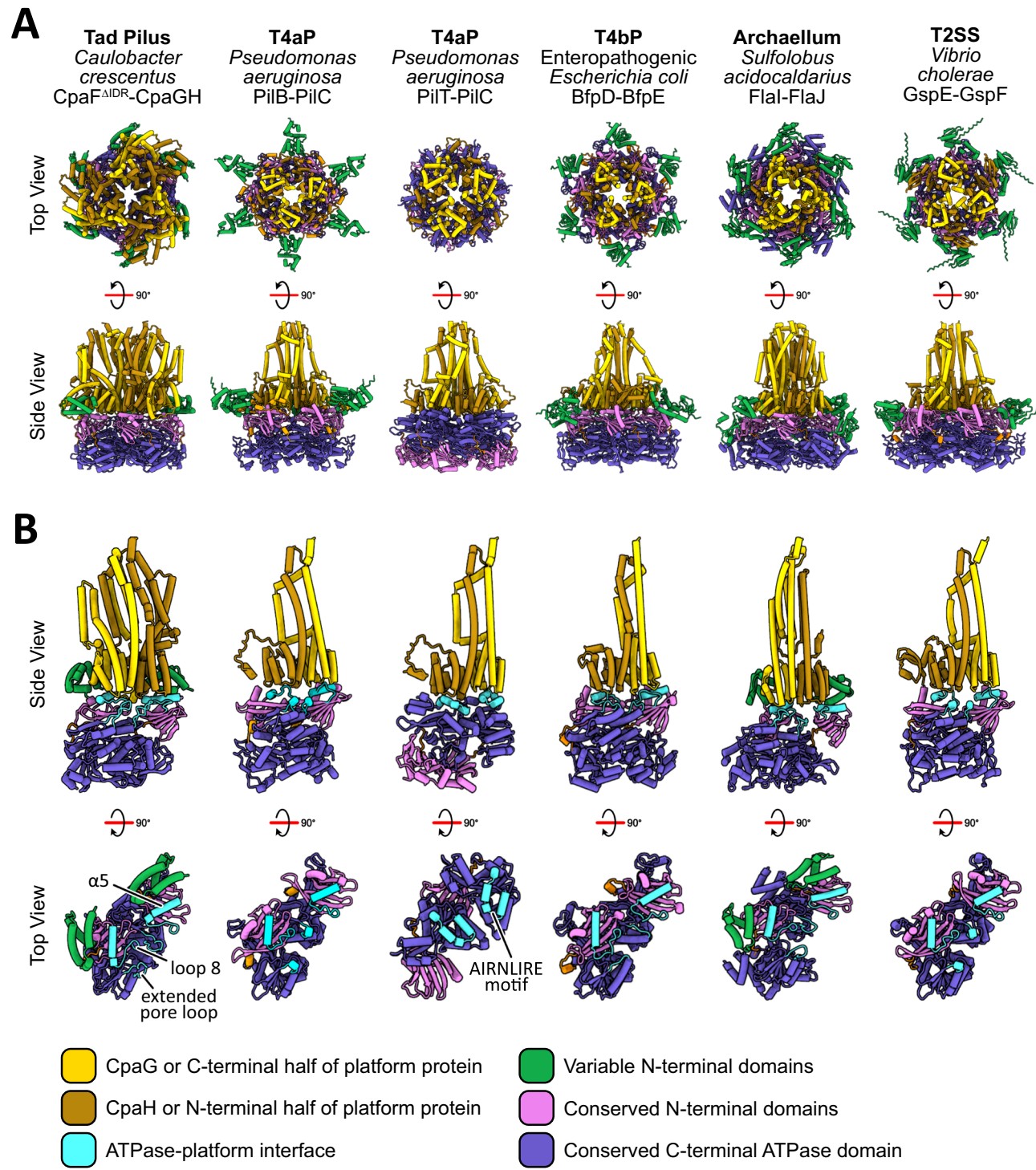

**Fig. 7 | AlphaFold3 modelling of motor subcomplexes from the TFF super-family. A** AlphaFold3 predicted motor subcomplexes from various TFF superfamily systems with the highest-ranking model depicted in both top and side views. The protein names and bacterial/archaeal species used to generate the predictions are listed in Supplementary Table 2. The Tad motor subcomplex was predicted with the major pilin PilA but was omitted from this analysis. ATPase models are colored according to the equivalent domain boundaries of CpaF. Platform models are colored by protein identity in the Tad pilus and by protein domains in every other TFF system. All ATPases are predicted with the conserved NTD contacting the platform proteins, with the exception of the PilT-PilC complex of the T4aP, which shows the ATPase in the opposite orientation. T4aP, type IVa pilus; T4bP, type IVb pilus; T2SS, type II secretion system. **B** Each platform protein, or in the case of the Tad motor, two platform proteins, interfaces with two monomers of the ATPase. The predicted binding interface for each system is colored in cyan. The ATPase-platform interface of the Tad motor comprises the α5 helix and its downstream loop 8, as well as the extended pore loop. The interface of the PilT-PilC complex is mediated by the conserved AIRNLIRE motif in PilT.

membrane into the central platform shaft through the openings between CpaGH heterodimers, gated by the N-terminal α1 of CpaH (Figs. 6A, E and 8). CpaF contraction to the compact state via ADP release is observed as a narrowing of the diameter of the hexamer pore relative to the expanded pore, suggesting this change could alter the conformation of the CpaGH platform complex (Figs. 3A and 8). This change could potentially impart mechanical forces to incorporate the extracted pilins into the filament. Thus, we hypothesize that

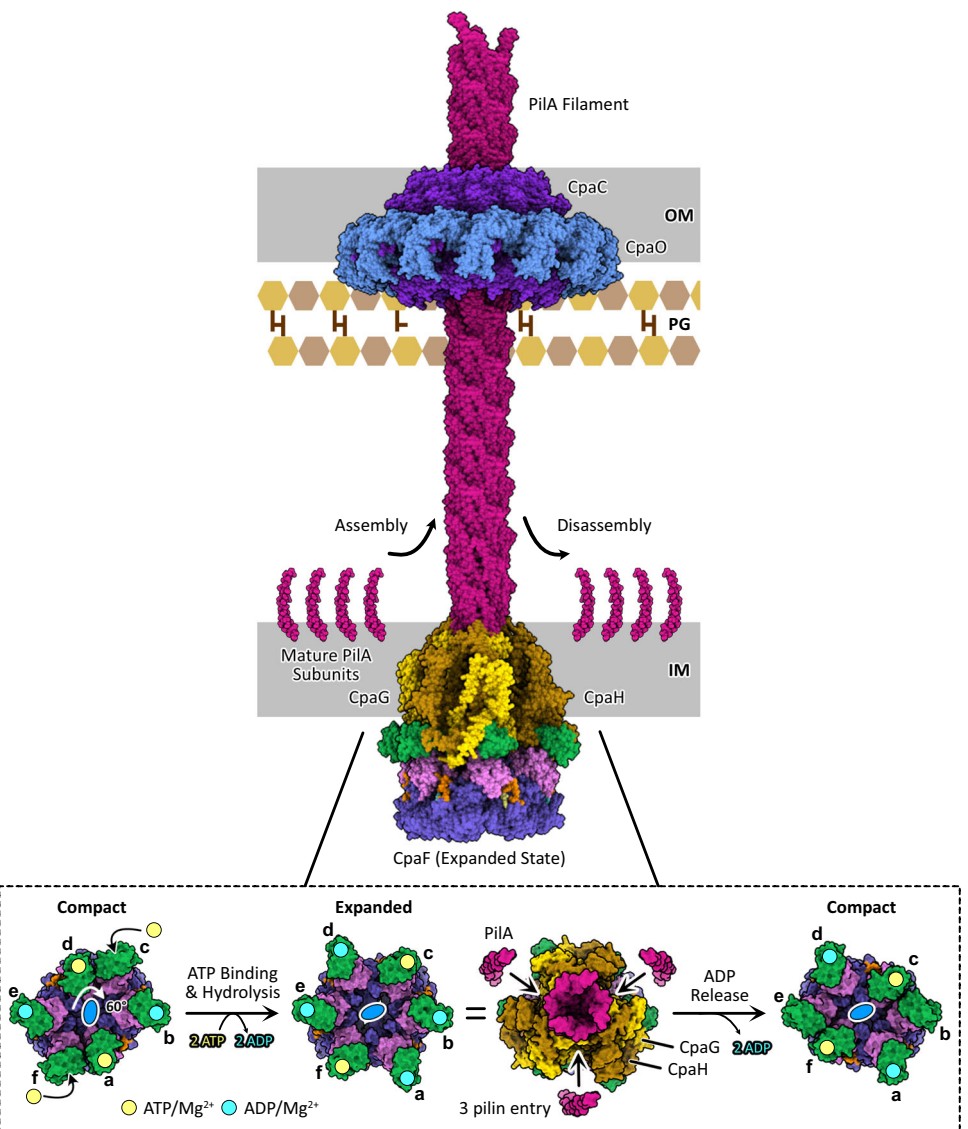

**Fig. 8 | Model of CpaF catalysis coupled to Tad pilus assembly.** (top) Schematic of the *C. crescentus* Tad pilus machinery and (bottom) the rotary model of CpaF catalysis. ATP binding and hydrolysis rotates the $C_2$ axis, and thus the platform protein complex, clockwise by 60°, converting CpaF from the compact to the expanded state. In this AlphaFold predicted conformation, CpaG and CpaH are positioned to facilitate pilin entry into the platform pore. Upon subsequent ADP release by CpaF and its transition to the compact state, the contracting mechanical forces are predicted to incorporate the pilins into the filament (PDB: 8U1K). Rounds of CpaF catalysis facilitate pilus assembly, which exits the outer membrane via the CpaC-CpaO secretin complex. The cryo-EM structure of the Tad secretin was determined from *Pseudomonas aeruginosa* (PDB: 8ODN).

nucleotide binding and hydrolysis expand and rotate the motor sub-complex to facilitate pilin subunit entry, while nucleotide release assembles pilins into the right-handed, helical filament[26] (Fig. 8). AlphaFold3 modeling predicts that all TFF motor subcomplexes involved in fiber extension employ a similar binding interface between their respective ATPases and platform proteins (Fig. 7), suggesting our model of Tad pilus assembly may be applicable to other TFF systems. An alternative rotary mechanism of CpaF catalysis using the sole CpaF$_{AMP-PNP}$ structure, in which nucleotide binding, hydrolysis, and release occur simultaneously, is proposed to assemble a right-handed, helical pilus[25] (Supplementary Fig. 15A).

The closed state of CpaF, harboring two ATP* and four Apo chains, is only observed in the absence of exogenous nucleotides, suggesting that this conformation likely does not participate in processive catalysis (Supplementary Fig. 9). In addition, the spatial position of the 3HBs in the closed state indicates that association with the platform proteins is unlikely. Polar CpaF foci are observed during periods of dynamic pilus activity, and disappear when pili are dormant, suggesting motor association with and dissociation from the machinery[22]. While this could indicate the existence of a paused, dissociated ATPase state, the closed structure is unlikely to represent this, given that the hexamer will need to release almost all its bound nucleotides. Thus, we hypothesize that this conformation could serve as a pre-assembly state, perhaps when the ATPase is trafficking to the poles. If the ATP* chains are bound to ATP molecules, then the binding of an additional pair of ATP molecules to the clockwise-adjacent Apo chains (chains a and d) would induce ATP binding and hydrolysis events to transition the hexamer into the compact state (Supplementary Fig. 16). At this point, the hexamer can engage the platform and initiate the catalytic cycle.

The $C_2$-symmetric structures of PilB and PilT support clockwise and counterclockwise rotation of PilC, respectively, during catalysis[18,19]. However, the sequential events of nucleotide turnover differ from CpaF (Supplementary Fig. 15B, C). In PilB and PilT, ATP binding and ADP

release are coupled to directionally opposite mechanical rotations, followed by ATP hydrolysis[18,19]. In PilT, ADP is not immediately released[19]. *C. crescentus* assembles Tad pili at a rate of ~750 Å/s[22], which given the helical rise of ~5 Å per PilA subunit[26], translates to an extension rate of 150 pilins per second. Considering our model proposes 2 ATP expenditures to assemble three pilins, roughly 100 ATPs will be consumed per second. *Pseudomonas aeruginosa* T4aP extends with an average rate of ~3600 Å/s[41] and rises by ~10 Å per major pilin[42]. This is equivalent to an assembly rate of 360 pilins per second, and assuming a three-pilin polymerization regime and 2 ATP consumption from the PilB structure[18], approximately 240 ATPs will be hydrolyzed per second. Our structures reveal that Tad and T4aP ATPases employ different catalytic sequences, despite operating via similar rotary mechanisms.

The mechanistic details of how CpaF retracts the Tad pilus remain unresolved. The nucleotide switch model proposes that differential affinity for ATP and ADP in the monomeric chains induces counter-clockwise motion, hence retraction, in CpaF[25]. Another possibility is that bidirectional rotation of the platform complex dictates pilus assembly versus disassembly, and by extension, the directionality of the ATPase. Our composite model of the Tad motor subcomplex suggests that each major pilin is contacted by a heterodimer of CpaGH (Fig. 6E). It is possible that a specific pilin interaction with CpaG or CpaH is correlated with extension or retraction and vice versa. A slippage in platform rotation, due to tension in the pilus fiber upon surface contact or interaction with other Tad components, or perhaps an unknown regulatory element that remains to be identified, could potentially alter the pilin-platform interaction, leading to directional switching. Our study offers a mechanistic framework for Tad pilus assembly, coupling CpaF catalysis to the clockwise mechanical rotation of CpaG and CpaH, while considerable work remains to unravel the details of Tad pilus disassembly.

## Methods
### Bacterial strains, plasmids, and growth conditions
Bacterial strains, plasmids, and primers used in this study are listed in Supplementary Table 3. *C. crescentus* strains were grown at 30 °C in peptone-yeast extract (PYE) medium[43]. PYE was supplemented with 5 µg/mL kanamycin (Kan), where appropriate, for plasmid maintenance. Commercially available, chemically competent *Escherichia coli* DH5a (NEB5α, New England Biolabs) was used for plasmid construction and was grown at 37 °C in lysogeny broth (LB) supplemented with 25 µg/mL Kan, where appropriate, for plasmid maintenance.

Plasmids were transferred to *C. crescentus* by electroporation, as described previously[44]. Chromosomal mutations were made by double homologous recombination using pNPTS138-derived plasmids, as previously described[45]. Briefly, plasmids were introduced into *C. crescentus* by electroporation, then two-step recombination was performed using Kan resistance to select for single crossovers, followed by sucrose resistance to identify plasmid excision events. All mutants were validated by Sanger sequencing using primers targeting outside the region of recombination to confirm the presence of the mutation.

For construction of the pNPTS138-derived plasmids, ~500-bp regions of DNA flanking either side of the desired mutation were amplified from *C. crescentus* NA1000 genomic DNA. Upstream regions were amplified using upF and upR primers, and downstream regions were amplified using downF and downR primers. upF and downR primers contained flanking sequences to facilitate insertion into pNPTS138 digested with EcoRV (New England Biolabs) by Gibson assembly (HiFi DNA Assembly Master Mix; New England Biolabs). Assembled plasmids were transformed into *E. coli* NEB5a (New England Biolabs), and clones with a positive insert were verified by Sanger sequencing.

Complementation constructs were made using pJC585, a reporter plasmid that contains RFP under the control of a taurine inducible promoter[46]. Expression from this promoter is leaky, so it is not necessary to add taurine to growth medium for gene expression. However, to maximize gene expression, growth medium was supplemented with 0.5 mM taurine, where necessary. The complete *cpaF* open reading frame, or the *cpaF* truncation variants, were amplified from *C. crescentus* NA1000 genomic DNA using comp-F and comp-R primers. comp-F primers encoded a synthetic ribosome binding site (TTTAAGAAGGAGATATACAT) and start codon, where necessary. The PCR fragments were digested with EcoRI and BamHI (New England Biolabs) and ligated into pJC585 digested with the same enzymes, which removes the vector-encoded rfp sequence. Assembled plasmids were transformed into *E. coli* NEB5a (New England Biolabs), and clones with a positive insert were verified by Sanger sequencing. To generate an empty pJC585 control vector that does not produce RFP, pJC585 was digested with KpnI, which removes ~250 bp of vector encoded *rfp* sequence, and re-ligated to generate pJC585-. Absence of RFP signal from *C. crescentus* strains carrying pJC585- was verified by fluorescence microscopy.

For construction of pET28a-derived plasmids, *cpaF* from *C. crescentus* NA1000 was PCR amplified from genomic DNA and cloned into pET28a with an N-terminal hexa-histidine tag using Gibson assembly (HiFi DNA Assembly Master Mix; New England Biolabs) to generate pET28a::*cpaF*. Truncation variants of CpaF were cloned into the same vector. pET28a::*cpaF*[K287A] was generated using around-the-horn PCR followed by Dpn1 digestion of the template vector. pET28a::*cpaF* was used as template for PCR amplification of the mutants. All plasmids were sequence verified by the TCAG sequencing facility (The Hospital for Sick Children, Canada).

### CpaF expression and purification
*Escherichia coli* Rosetta 2 cells were transformed with pET28a::*cpaF* or mutants of *cpaF*. Cells were grown in 2 L of LB at 37 °C supplemented with 50 µg/mL Kan to an $OD_{600}$ of 0.6–0.8. Protein expression was induced by the addition of 0.5 mM isopropyl-D-1-thiogalactopyranoside for 16 h at 18 °C. Cells were harvested by centrifugation at $7000 \times g$ for 15 min. Pellets were resuspended in 30 mL of binding buffer (50 mM HEPES, pH 8, 500 mM NaCl, 10% (v/v) glycerol, 50 mM imidazole, 1 mM TCEP) in addition to a crushed protease inhibitor cocktail tablet (SIGMA*FAST*™, EDTA-free), 1 mM phenylmethylsulfonyl fluoride, and small amounts of powdered DNase I (Bio Basic) and lysozyme (Bio Basic) to aid in DNA fragmentation and cell wall disruption, respectively. Upon cell lysis by passage through an Emulsiflex-c3 homogenizer, insoluble cell debris was clarified by centrifugation at $35,000 \times g$ for 40 min. The supernatant was filtered and incubated with 2.5 mL of Ni-NTA agarose resin (Qiagen) pre-equilibrated with binding buffer, followed by a 75 mL-wash with the same buffer prior to elution in 25 mL of elution buffer (binding buffer plus 500 mM imidazole). The sample was concentrated using an Amicon Ultra-15 centrifugal device (Millipore) with 100 kDa molecular weight cut-off and subsequently injected into a Superose 6 increase 10/300 GL size exclusion column (GE Healthcare) pre-equilibrated in SEC buffer (50 mM HEPES, pH 8.0, 200 mM NaCl, 10% (v/v) glycerol, 1 mM TCEP). Fractions containing purified CpaF were assessed by SDS-PAGE, concentrated, and flash frozen in liquid nitrogen for storage at −80 °C.

### Cryo-EM sample preparation
Nanofabricated holey gold grids containing regular arrays of approximately 2-µm holes were prepared in-house as previously described[47]. Immediately prior to freezing, 10% (v/v) glycerol in the protein buffer was removed with a 40 kDa molecular weight cut-off Zeba Spin desalting column (ThermoFisher) equilibrated with freezing buffer (50 mM HEPES, pH 8.0, 200 mM NaCl, 1 mM TCEP). 5 mg/mL of freshly purified CpaF was supplemented with a final concentration of 0.05% (w/v) CHAPS (BioShop) to improve particle orientations. The under-saturated ATP/ADP sample was prepared by mixing a final

concentration of 50 µM of an equimolar ATP/ADP/MgCl$_2$ mixture in freezing buffer with CpaF and incubating on ice for 10 min. A similar approach was used to prepare the saturated ATP/MgCl$_2$ sample to a final concentration of 1 mM. Grids were glow-discharged in air for 2 min prior to a 2 µL sample application, followed by blotting in a Leica EM GP2 plunge freezer (Leica Microsystems) for 1.5 s at 4 °C and 95% relative humidity before plunging into liquid ethane.

## Cryo-EM data collection

All samples were screened with a Glacios electron microscope (ThermoFisher) operating at 200 kV and equipped with a Falcon 4i camera. Movies were collected with a defocus range of 1 to 2 µm in electron event representation (EER) format[48] using the EPU software. A nominal magnification of 92,000×, corresponding to a calibrated pixel size of 1.5 Å per pixel was used. The exposure dose was ~40 electrons/Å$^2$. Screened autogrids with a sufficient distribution of particle orientations and thin ice gradient were directly used for high-resolution data collection on the Titan Krios G3 electron microscope (ThermoFisher) operating at 300 kV and equipped with a Falcon 4i camera. Automated data collection was performed using EPU with aberration-free image shift and fringe-free imaging. Movies were collected with a defocus range of 1 to 2 µm in EER format at a nominal magnification of 75,000×, equivalent to a calibrated pixel size of 1.03 Å per pixel. The camera exposure rate and the total exposure for the first apo and under-saturated ATP/ADP datasets were 5.2 electrons/pixel/s and ~36.7 electrons/Å$^2$, respectively. Movies for the under-saturated ATP/ADP dataset were additionally collected at 30° and 35° stage tilts with the same microscope parameters as above. Exposure rate and total exposure for the second apo and saturated ATP datasets were 6.8 electrons/pixel/s and ~42 electrons/Å$^2$, respectively.

## Cryo-EM image analysis

All image analysis was performed with cryoSPARC[49] v.4.2.2 and v.4.4.0. Movie frames were aligned with patch motion correction and contrast transfer function (CTF) parameters were estimated in patches using cryoSPARC live[50]. Movies with CTF fit estimates above 6 Å and a defocus range below 0.5 µm and above 3.0 µm were rejected. Templates for particle picking were generated by 2D classification of blob picked particle images. Template picked particles were separated into top/bottom and side/tilt views by 2D classification for Topaz[51] training and further particle picking. Duplicate particle images from different picking strategies were removed and binned to 128 × 128 pixel boxes, totaling 2,794,604, 2,078,699, and 2,250,189 particle images for the apo, under-saturated ATP/ADP, and saturated ATP datasets, respectively. Particles were subjected to rounds of ab initio 3D classification[49] and heterogenous refinement to exclude dominant top view and junk particles. Retained particles were local motion corrected[52], re-extracted with 288 × 288 pixel boxes, and further subjected to one round of *ab initio* 3D classification and heterogeneous refinement. Classes were refined using non-uniform refinement[53] with C$_2$ symmetry enforced to generate hexameric consensus maps of the closed state (84,636 particle images, 3.5 Å resolution) and compact state (149,839 particle images, 3.2 Å resolution) from the apo dataset, compact state (226,031 particle images, 2.9 Å resolution) and expanded state (138,569 particle images, 3.3 Å resolution) from the under-saturated ATP/ADP dataset, and finally another expanded state (273,260 particle images, 3.3 Å resolution) from the saturated ATP dataset. Reference-based motion correction and subsequent CTF refinement were applied to the particle images of the closed state and the two structures from the under-saturated ATP/ADP dataset, improving their respective global resolutions to 3.4, 2.8, and 3.2 Å, respectively. All consensus maps were locally sharpened with LocalDeblur[54], embedded within Scipion[55] v.3.3.0. To improve the map density of the 3HB region, local refinement with Gaussian prior enabled was performed using C$_2$ symmetry expanded particles and a binary mask covering the 3HBs

and NTDs of the asymmetric trimer. The resolution range of all five focus refined maps was between 3.2 and 4.1 Å, though particle alignment from the NTD likely overestimated these values. The improved 3HB densities were duplicated to the other asymmetric unit and combined with the consensus map in UCSF ChimeraX[56] v.1.7 via the vop maximum function to generate composite maps.[57]

## Model building and refinement

The resolution of all five maps was sufficient for model building. For each structure, six copies of Alphafold2 predicted CpaF monomers were rigid body fitted into the consensus map in UCSF ChimeraX. A combination of the locally sharpened and composite maps was employed to manually adjust residues in Coot[57] v.0.9.8.7. Dihedral angles, rotamers, and Ramachandran outliers were further fixed in ISOLDE[58] v.1.6 before subjecting models to Phenix real space refinement[59] v.1.20.1-4487. In all five atomic models, residues 147–501 of NTD-CAD were built, while residues 80–146 of the 3HB were modeled without side chains. All structural figures and movies were generated using UCSF ChimeraX.

## Phylogenetic analysis

A previously cultivated list of CpaF protein sequences was used[22]. From this initial list, groups of sequences with >80% pairwise identity were reduced to a single representative sequence. The remaining sequences were queried against the Alphafold Protein Structure Database[30] (AFDB) to retrieve predicted structures of each CpaF ortholog. Any sequence that did not have a structure in the AFDB, or a structure of an orthologous protein with >94% sequence identity, was eliminated from the list. This left 292 CpaF sequences, whose accession numbers can be found in Supplementary Data 1, that were used for phylogenetic analysis, as well as three archaellar ATPases that were used as an outgroup. These sequences were aligned using the default parameters of Muscle (version 5.1, as implemented in Geneious Prime 2024.0.7). The resulting alignment was used to generate a phylogenetic tree with RAxML (version 8.2.7, as implemented in Geneious Prime 2024.0.7) set to perform 100 rapid bootstraps and subsequent maximum-likelihood search using the GAMMA model of rate heterogeneity and JTT substitution model[60]. The resulting tree was visualized using the Interactive Tree of Life software[61] (iTOL) and can also be viewed at https://itol.embl.de/tree/1322042512522045617219208066. The root was inferred as the branch separating the archaellar ATPases from the CpaF-like ATPases. The predicted structure of each CpaF ortholog retrieved from the AFDB was used to assign each sequence to one of three architectural groups, based on the presence or absence of specific domains, and these groupings were mapped onto the phylogenetic tree using iTOL.

## Enzyme-coupled ATPase assay

To assess CpaF activity, an enzyme-coupled ATPase assay was performed. Briefly, 2.5 µg of WT or mutant CpaF constructs was incubated in a reaction buffer containing 0.84 U lactate dehydrogenase (Sigma), 0.6 U pyruvate kinase (Sigma), 50 mM HEPES, pH 8.0, 200 mM NaCl, 5% (v/v) glycerol, 1 mM TCEP, 1 mM MgCl$_2$, 0.8 mM nicotinamide adenine dinucleotide (NADH) (Bioshop), 1 mM phosphoenolpyruvate (Sigma), and 1 mM ATP (Bioshop). The total reaction volume was 120 µL. Reactions, corresponding to the conversion of NADH to NAD$^+$, were performed in triplicate and measured at an absorbance of 340 nm every minute for 1 h at 30 °C. Background signal without CpaF was subtracted. The linear slope between 30 and 40 mins, along with an NADH standard curve, was used to calculate specific activity.

## Thermal shift assay

Protein thermal stability was assessed through heat denaturation using a thermocycler. 10 µg of WT or mutant CpaF constructs was added to a 25 µL reaction, along with a final concentration of 5× SYPRO orange

stain (Invitrogen) and SEC buffer. Temperature was increased every 30 s in 0.5 °C increments from 10 to 95 °C and the HEX channel was used to measure and generate fluorescent melt curves. Melting temperatures were calculated at the inflection point of each curve.

## Pilus labeling, blocking, imaging, and quantification

The pili of *C. crescentus* were labelled as described previously[33]. Briefly, 25 μg/mL of Alexa Fluor 488 $C_5$ Maleimide (AF488-mal, ThermoFisher Scientific) was added to 100 μL of early exponential phase *C. crescentus* cell culture ($OD_{600}$ = 0.1–0.3) and incubated for 5 min at room temperature. To artificially block pilus retraction, 500 μM of methoxy-polyethylene glycol maleimide with an average molecular weight of 5 kDa (PEG5000-mal, Sigma) was added to the *C. crescentus* cell culture immediately prior to the addition of 25 μg/mL AF488-mal. Labelled and/or blocked cells were collected by centrifugation for 1 min at 5000 × $g$ and washed once with 100 μL of PYE to remove excess dye. The cell pellets were resuspended in 20 μL of PYE, 1 μL of which was spotted onto a 1% agarose PYE pad (SeaKem LE, Lonza Bioscience). The agarose pad was sandwiched between glass coverslips for imaging, which was performed using a Nikon Ti-2 inverted fluorescence microscope with a Plan Apo 60× objective, a green fluorescent protein (GFP) filter cube, a Hamamatsu OrcaFlash 4.0 CCD camera, and Nikon NIS Elements imaging software. The percentage of cells within the population with fluorescent cell bodies, and the percentage of cells with blocked and labelled pili, were quantified manually using ImageJ software[62].

## Western blot analysis

To determine the amount of CpaF produced by different *C. crescentus* strains, approximately $10^9$ cells from early exponential phase cultures ($OD_{600}$ = 0.1–0.3) were collected by centrifugation for 5 min at 5000 × $g$. The supernatant was removed, and the cell pellets were resuspended in 100 μL of 4× SDS loading buffer (200 mM Tris-HCl pH 6.8, 40% (v/v) glycerol, 8% (w/v) SDS, 20% (v/v) β-mercaptoethanol, and 0.005% (w/v) bromophenol blue). Samples were boiled for 10 min and then separated by SDS-PAGE using 12% gels. The samples were transferred to nitrocellulose membranes (100 V, 1.5 h) and blocked in 5% (w/v) skim milk powder resuspended in Tris buffered saline with Tween-20 (TBS-T; 10 mM Tris-HCl pH 7.5, 150 mM NaCl, 0.05% (v/v) Tween-20) for 2–4 h. Membranes were probed with α-CpaF antibodies[22] (Biomatik) at 1:5000 dilution in 1% (w/v) skim milk powder resuspended in TBS-T for 16 to 20 h. Membranes were then washed four times with TBS-T and probed with horseradish peroxidase-conjugated goat anti-rabbit antibody (Pierce, #1858415) at 1:5000 dilution in 1% (w/v) skim milk powder resuspended in TBS-T for 1 h. Membranes were washed again four times with TBS-T, developed using SuperSignal West Pico Plus chemiluminescent substrate (ThermoFisher Scientific), and imaged on a Bio-Rad ChemiDoc MP imaging system. As a loading control, membranes were probed with α-GAPDH antibodies (Biomatik, produced using full-length *C. crescentus* GAPDH in New Zealand White rabbits) at 1:5000 dilution, as described above. The Western blot images shown are representative of three independent biological replicates.

## AlphaFold3 modeling

Motor subcomplex predictions from all TFF systems were generated using the AlphaFold3 server[34] (https://alphafoldserver.com). Bacterial species and protein identities were retrieved from Uniprot and are listed in Supplementary Table 2. Six copies of the ATPase monomer and three copies of the platform protein monomers were provided as input for each complex prediction. An additional seven copies of the pilin subunit, as well as two ATPs, four ADPs, and six $Mg^{2+}$ ions were used for the Tad motor subcomplex prediction. All predictions were additionally colored by their pLDDT score in ChimeraX.

## Reporting summary

Further information on research design is available in the Nature Portfolio Reporting Summary linked to this article.

## Data availability

Cryo-EM consensus, locally refined, and composite maps are deposited, respectively, in the Electron Microscopy Data Bank under accession codes EMD-47431, EMD-47432, EMD-47433 (Closed-Apo), EMD-47434, EMD-47435, EMD-47436 (Compact-Apo), EMD-47437, EMD-47438, EMD-47439 (Compact-Undersaturated-ATP/ADP), EMD-47440, EMD-47441, EMD-47442 (Expanded-Undersaturated-ATP/ADP), and EMD-47444, EMD-47445, and EMD-47446 (Expanded-Saturated-ATP). The corresponding atomic models are deposited in the Protein Data Bank under accession codes 9E24 (Closed-Apo), 9E25 (Compact-Apo), 9E26 (Compact-Undersaturated-ATP/ADP), 9E27 (Expanded-Undersaturated-ATP/ADP), and 9E29 (Expanded-Saturated-ATP). PDB codes of previously published structures used in this study are 8U1K and 8ODN. Strains and plasmids are available from the corresponding authors upon request. Source data are provided as a Source Data file. Source data are provided with this paper.

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

## Acknowledgements

We thank Deepa Raju and members of the Howell lab for insightful discussions during manuscript preparation. We thank Gautier Courbon, Yingke Liang, Hanlin Wang, Hui Guo, Justin Di Trani, Samir Benlekbir and Zhijie Li for technical assistance with cryo-EM grid freezing, data collection, and image processing. We thank Jun Liu for the helpful discussion comments on the manuscript. We appreciate Rhett A. Snyder for generation of strain YB8446 and the pNPTS138::ΔcpaF plasmid. I.Y.Y. was supported by a Natural Sciences and Engineering Research Council PGS-D scholarship and a Hospital for Sick Children Research Institute Restracomp scholarship. G.B.W. was supported by a Fonds de Recherche du Québec – Nature et Technologies postdoctoral fellowship. This work was funded by project grant PJT-169053 from the Canadian Institutes of Health Research to L.L.B., Y.V.B. and P.L.H. J.L.R. is the recipient of a Tier I Canada Research Chair in Electron Cryomicroscopy. L.L.B. holds a Tier I Canada Research Chair in Microbe-Surface Interactions. Y.V.B. is a Canada 150 Research Chair in Bacterial Cell Biology. P.L.H. was the recipient of a Tier I Canada Research Chair in Structural Biology from 2006–2020. Cryo-EM data were collected at the Toronto High-Resolution High-Throughput Cryo-EM facility, supported by the Canada Foundation for Innovation and Ontario Research Fund. In vitro assays were performed at the Structural & Biophysical Core (SBC) facility at The Hospital for Sick Children.

## Author contributions

I.Y.Y. cloned, expressed, and purified all CpaF constructs for cryo-EM and in vitro assays, prepared cryo-EM samples, imaged specimens, processed images, built all atomic models, and performed all structural analyses and predictions. G.B.W. performed in vivo imaging of CpaF IDR mutant cells and the phylogenetic study. J.L.R. provided mentorship and guidance on cryo-EM sample preparation and image processing. I.Y.Y. and G.B.W. wrote the manuscript and prepared figures. L.L.B., Y.V.B. and P.L.H. conceived, supervised, and coordinated the project. All authors contributed to the editing of the manuscript.

## Competing interests

All authors declare no competing interests.
