## [Transparent Peer Review file · Nature Communications]

Conformational changes in the motor ATPase CpaF facilitate a rotary mechanism of Tad pilus assembly

Corresponding Author: Dr P. Lynne Howell

Version 0:

Reviewer comments:

Reviewer #1

(Remarks to the Author)

Review of Yen et al, Coordinated conformational changes in the Tad pilus ATPase CpaF facilitate a rotary mechanism of catalysis for Nature Communications

This is a tour-de-force manuscript reporting the high-resolution single particle cryo-EM reconstructions of five independent nucleotide-bound states of the *C. crescentus* Tad pilus assembly (and retraction) ATPase CpaF at nominal resolutions of 2.8 – 3.4 Å. Additionally, phylogenetic analysis of CpaF structural classes, mutagenesis and subsequent analysis of the role of the CpaF intrinsically disordered region, and substantial AlphaFold 3 modeling add to the experimental results. Combined with recent PNAS publication from Craig/Wright/Egelman groups on the structure of PilA, the field thus has experimental structural information about the pilin subunit and its assembly motor for the very widespread and ancient TadA system.

One of the novel insights comes from the phylogenetic analysis for CpaF, showing that most species lack the IDR in their CpaF-like motors (Class I). Thus the IDR is not intrinsically required for an in-and-out motor function. Nonetheless, its deletion can impact both protein stability and apparently regulation of retraction AND assembly.

Hohl, Banks, Manly, Le and Low published cryoEM structures of different nucleotide-bound states of *C. crescentus* CpaF very recently in this same journal. There can certainly be value in two groups carrying out detailed analysis of the same molecule; the approaches and interpretations and biases leading to deeper and broader understanding of a phenomenon. Hohl et al. is respectfully cited but not until the Discussion. Two main discrepancies are discussed, namely the difference in nucleotide bound states (potentially due to the presence of HEPES) and the role of the IDRs.

Overall, the model is that expansion from a compact state allows nucleotide triphosphate binding and hydrolysis during a 60 degree axis rotation, while contraction then allows ADP release and the return to the compact state for subsequent nucleotide triphosphate binding.

Specific suggestions;

1. Abstract and Introduction are set up to suggest the paper will explain the bidirectionality of CpaF. But the experimental results are about assembly with a minor mention of switching in the discussion. Please don't imply in the abstract that bidirectionality mechanism will be resolved.
2. Please present the main conclusions in abstract; currently it is a self-evident and incontrovertible paragraph. How is the understanding of the mechanism of assembly, or the mechanism of switching between assembly and retraction, more well understood with this addition to the oeuvre?
3. While there is no question the authors have provided additional structures and novel insights into the Type IV pilus assembly motors, they lay claim to more than they should. While this is not a review article and one does not expect a fulminant review of all the T4P motor protein structures and their conclusions, a few more acknowledgements of what has come before and has added to our understanding would be appropriate, in addition to the two cited McCallum papers. The C2 packing, and three states of the nucleotide binding pocket have been reported before. Some potential sources are:

Solanki, Kapoor and Thakur; FEBS J 2018 doi: 10.1111/febs.14619

Nayak, Singh, Zhao, Samsó, Donnenberg mBio 2022 doi: 10.1128/mbio.02270-22

4. The Interchain contact that mediates nucleotide binding eg the Arginine Fingers is nicely described. But see Satyshur et al. 2007 description of conserved arginines in adjacent subunit and mutation of conserved LRAAL*R*E (R347 equivalent) that prevents retraction when installed in *P. aeruginosa* PilT at equivalent R194 (re Suppl. Fig 10b). This 2007 paper already describes the interchain contact, shows low resolution electron density of *A. aeolicus* PilT Arg 207 with modeled PO4 (Figure 6) and points out, "A cluster of acidic residues is also found near bound nucleotide at the subunit interface in structures of the VirB11 ATPase HP0525, in which it was proposed that ATP prevents the electrostatic clash among the arginines, thus allowing domain closure (Yeo et al., 2000). Interestingly, only Arg207 is invariant; the specific position of neighboring arginines is different in PilT versus the VirB11 ATPases (Figure 1). To our knowledge, this is the first time the arginine finger role has been suggested in PilT or any other member of the type II/type IV secretion ATPase family. Although our results do not permit us to draw detailed conclusions about the role of Arg207 in subunit interactions, they do provide testable hypotheses regarding the roles of the arginine wire in pilus retraction." This paper also previews Yen et al. by suggesting, "that an arginine wire may lead the released Pi product out of the active site".

5. AlphaFold multimer modeling including the Pila pilin subunits is bold. But, the motor protein came back as a C6 hexamer and the pilin subunits did not match the cryoEM structure of Sonani et al. (Figure 6A, left side). For this reason, the authors very reasonably created a composite model based on experimentally-determined structures. Thus, Panel A, left side should be left out altogether.

6. The Alpha fold structures which are the most reliable (all blue in Suppl Figure 13) presumably are so because there are deposited crystal structures included in the training set. Ideally these should be cited and/or should this point should be explicit when comparing the alpha fold models.

7. The model for retraction is proposed to be a transition from Compact hexamer to Expanded hexamer and back to Compact hexamer, as seen in Figure 3 (which is busy; can the many vector arrow structures on the top line of Panel B be left out?) and Figure 8 (which is fantastic). During the first transition, two subunits which are "Compact-Apo" bind to ATP to form Exp-ATP while two subunits which are bound to ATP (Com-ATP) hydrolyze to become Exp-ADP. Then in the transition back to Compact, the original two ADP sites release nucleotide to become Com-Apo.

"Closed, compact and expanded" hexamers are seen, with closed only observed in the absence of nucleotide. The biological role if any of the closed state was not clear. Is the "Closed" hexameric CpaF not part of the normal in-vivo rotary motor trajectory for pilus assembly, serving as a paused state in the absence of ATP or perhaps during a regulated phase when assembly is halted by interaction with another protein?

8. The idea of the supplemental movies is great, but only the full hexamer view turns out to be helpful. The others need less blurry monomer drawings; perhaps less wide cartoon elements.

9. Because alpha 5, loop 8 and extended pore loop are colored light blue in Figure 6B but not shown in Figure 1A, I have a hard time understanding if these elements are within the 3HB itself or within the NTD. Please clarify, perhaps by coloring in Figure 1A, or by listing the amino acid ranges of these secondary structure elements in the text or the legend to Figure 6.

Minor comments:

1. Reference 18, Sonani et al., some names are misspelled.

Reviewer #2

(Remarks to the Author)

In this manuscript, Yen and co-workers report a structural and functional characterization of the Tad pilus ATPase CpaF, in the bacterium *Caulobacter crescentus*. Strikingly, they have obtained three distinct conformations for the CpaF hexamer, which correlates with different nucleotide states, and allows the authors to propose a complete molecular mechanism for CpaF-mediated filament assembly, supported by very elegant structure-function analyses.

This study forms a significant advance in our understanding of the Tad pilus, a bacterial surface appendage of increasing interest due to its wide-spread prevalence in bacteria, and indeed of the related T4P and archaellum. It is well suited for publication in Nature Communications.

Of course, the "elephant in the room" here is the study from Hohl et al, published a few months ago, and which also reports the structure of CpaF. The authors largely ignore this study, and only mention it in one paragraph, in the discussion section. While we sympathize with the authors' attempt to conserve the novelty of their work, considering that Hohl et al. was published > 4 months ago, ignoring it rather detracts from the message of this current work, and indeed the study reported here forms significant advance from the data reported in Hohl et al. Therefore our main recommendation would be to re-write several sections of the manuscript to take this into account. Notably, we recommend the following changes:

- The introduction should mention the results from Hohl et al, and highlight the outstanding questions that resulted from it (most strikingly, the single conformation state of the CpaF hexamer).

- Supplementary figures 1C and D, as well as 8C, are critical to understand the conformation distribution, and the role of related nucleotide states. These should be part of a main figure.
- Similarly, the structure-function analyses (figure 5) are critical, and the corresponding data (Figure S11A/B, S12B/C) should be included in the main figures.
- In contrast, the phylogenetic analyses (Figure 4 and 7) do not really contribute much to this work, and rather distract from its main message. These should be relegated to the supplementary material, and the corresponding results section should be shortened.

Other major comments that need to be addressed:

- The authors go to great lengths to resolve the conformation of the 3HB domain in their maps, and indeed its conformation varies very significantly between the different conformations. Yet it is not involved in the interaction with the nucleotide, and indeed the modelling suggests that its role may be to interact with CpaG and CpaH (lines 364-370). A more in-depth analysis of the different positions observed for this domain, how it may be affected by the conformation of the hexamer, and how it may affect interaction with the assembly machinery and/or the pilin protein, would be beneficial.
- The ATPase activity assay reported in Figure 5 is important as it establishes that the CpaF hexamer is functional in isolation, validating the different conformations observed. However this experiment requires a negative control, with a mutant lacking catalytic activity. Ideally, a positive control, with another well-characterized ATPase, could also be used.

Additional minor comments:

- Line 27: As powerful as AF3 is, we don't think it is accurate to say that it "demonstrates" something, but rather suggest/indicates.
- Line 35: we find the term TFF rather confusing, and although it has been employed in Denise et al (PloS Biol, 2019),
- Line 101: What is the rationale for using ADP and ATP at under-saturating condition? Obviously it produced a very interesting result, but is this a common/previously used strategy to characterize AAA+ ATPases? If so, can the authors provide references?
- Lines 338-339, and 344-346: AF3 cannot model helical assemblies, therefore including PilA here was not going to work. This should not be included.
- Lines 502-505: An additional figure, perhaps as part of Figure 8, comparing the mechanisms of CpaF to that of PilB and PilT, would help illustrate this.
- Lines 99, 106, 142: The authors use unsharpened maps to analyse nucleotide density. Considering the authors' expertise, we have no doubt that it is the correct thing to do – however this is not standard procedure these days. We encourage the authors to add a sentence or two to explain the rationale here, with reference if appropriate.

Reviewer #3

(Remarks to the Author)

Reviewer #4

(Remarks to the Author)

This review has been written with the help of a graduate student for training purpose with approval by the editor.

The manuscript by Yen et al. analyzes the structure of the pilus motor complex CpaF in *C. crescentus* by cryoEM and how nucleotide exchange yields conformational changes that drive function. CpaF is a member of the single motor type IV filament superfamily (TFF), which facilitates extension and retraction of type IV pilus (TFP) filaments. This is in contrast with other members of this family that have dedicated extension and retraction ATPases. The process of how this single motor facilitates both directions is unknown, and the current manuscript presents important insights into this question. The manuscript has done an excellent job thoroughly explaining CpaF structure and mechanism of rotation in a way that is easy to follow and well written. The diverse combination of experimental data, structural analysis, dynamics, and modeling implemented by the authors is key for understanding how these complexes function in-vivo and interact with the rest of the TFP machines.

We are not experts in protein structure or cryoEM and leave judgement of the details to other reviewers, but the results and conclusions appear technically sound to us. We are experts for functional in-vivo assays of TFP dynamics as shown Fig. 5. These data are a strength of this manuscript and aid to the structural data (structure papers are often lacking experimental connections to in-vivo). As the authors point out, their findings and proposed model for how this motor class drives TFP extension/retraction has broad implications for the large field of TFF systems.

We would like to point out that the current manuscript adds much in rigor, depth, and quality compared to reference 31, supported by more detailed experiments that yield better insights and understanding of how the CpaF system might work. We only have several minor comments and recommend publication after these have been addressed.

Minor comments:

- The manuscript refers to "PilT-like structure" several times. It is unclear why not compare to a PilB-like structure?
- cryoEM is used as abbreviation for "electron cryo-microscopy", which seems to be a mix-up in order of words in the abstract and introduction
- Some language, especially in the beginning of the results, uses jargon that is not easy to follow for non cryoEM or structure

experts. Please explain some terminology, for example C2 symmetry, what are particles (especially, are these monomers or oligomers), the reference of chains a and d (line 95-96), what does over-refined mean (line 123).

- Line 178 – “walker A G284” – this is the only thing that is not labeled on the graph it would be good to add it in

- Lines 233-252: This description was challenging to follow. Please consider rephrasing.

- Lines 253-260: If CpaF always rotates in the same direction, how can it facilitate extension and retraction?

- Figure 3D: the coloring of the in-figure legend is ambiguous. Please either change colors or add corresponding bars with/without patterns in front of each item in the legend.

- Fig 7A: The docking of N/C terminal domains seems to be different between the different systems. Can the authors elaborate?

- Since one of the main premises of the paper is to understand how a single motor can drive extension and retraction, can the authors elaborate more on this point? Could it be that CpaG and CpaH each mediate one direction and are exchanged between extension/retraction?

Reviewer #5

(Remarks to the Author)

Version 1:

Reviewer comments:

Reviewer #1

(Remarks to the Author)

The authors have done a thorough job responding appropriately to all of the reviewers' comments.

Reviewer #2

(Remarks to the Author)

The authors have extensively addressed the comments from all three reviewers, and as a consequence the revised manuscript is a lot stronger. We are now delighted to recommend its publication in Nature Communications.

If possible, we would recommend the following minor adjustment:

- We agree with the authors' suggestion not to include the PilT control in the main figure, as the K287A mutant is a very strong negative control, which very elegantly support their structure. However, the fact that CpaF is a lot more active than PilT, in isolation, is a very interesting observation, which could be related to the bi-directional role of CpaF? If there is enough room, I would recommend that the authors add this as a panel in the supplementary material, and include a sentence mentioning this.

Reviewer #3

(Remarks to the Author)

Reviewer #4

(Remarks to the Author)

The authors have addressed all of our concerns.

Reviewer #5

(Remarks to the Author)

RESPONSE TO REVIEWER'S COMMENTS:

Our responses below are in blue.

Reviewer #1 (Remarks to the Author):

Review of Yen *et al*, “Coordinated conformational changes in the Tad pilus ATPase CpaF facilitate a rotary mechanism of catalysis” for Nature Communications.

This is a tour-de-force manuscript reporting the high-resolution single particle cryo-EM reconstructions of five independent nucleotide-bound states of the *C. crescentus* Tad pilus assembly (and retraction) ATPase CpaF at nominal resolutions of 2.8 – 3.4 Å. Additionally, phylogenetic analysis of CpaF structural classes, mutagenesis and subsequent analysis of the role of the CpaF intrinsically disordered region, and substantial AlphaFold 3 modeling add to the experimental results. Combined with recent PNAS publication from Craig/Wright/Egelman groups on the structure of PilA, the field thus has experimental structural information about the pilin subunit and its assembly motor for the very widespread and ancient TadA system.

One of the novel insights comes from the phylogenetic analysis for CpaF, showing that most species lack the IDR in their CpaF-like motors (Class I). Thus, the IDR is not intrinsically required for an in-and-out motor function. Nonetheless, its deletion can impact both protein stability and apparently regulation of retraction AND assembly.

Hohl, Banks, Manly, Le and Low published cryoEM structures of different nucleotide-bound states of *C. crescentus* CpaF very recently in this same journal. There can certainly be value in two groups carrying out detailed analysis of the same molecule; the approaches and interpretations and biases leading to deeper and broader understanding of a phenomenon. Hohl *et al*. is respectfully cited but not until the Discussion. Two main discrepancies are discussed, namely the difference in nucleotide bound states (potentially due to the presence of HEPES) and the role of the IDRs.

Overall, the model is that expansion from a compact state allows nucleotide triphosphate binding and hydrolysis during a 60-degree axis rotation, while contraction then allows ADP release and the return to the compact state for subsequent nucleotide triphosphate binding.

We thank the reviewer for their positive comments and for acknowledging the value that our manuscript adds to the understanding of CpaF structure and function in addition to the foundation established by Hohl *et al*¹. We now refer to and describe the Hohl *et al* study in the introduction (lines 77-82), which will more clearly delineate how we build upon their findings.

Specific suggestions:

1. Abstract and Introduction are set up to suggest the paper will explain the bidirectionality of CpaF. But the experimental results are about assembly with a minor mention of switching in the discussion. Please don't imply in the abstract that bidirectionality mechanism will be resolved.

We recognize that our experimental findings do not resolve the bidirectional mechanism of CpaF and thus have removed these implications from the abstract and introduction.

2. Please present the main conclusions in abstract; currently it is a self-evident and incontrovertible paragraph. How is the understanding of the mechanism of assembly, or the mechanism of switching between assembly and retraction, more well understood with this addition to the oeuvre?

We have completely re-written the abstract to better summarize our main findings and conclusions.

3. While there is no question the authors have provided additional structures and novel insights into the Type IV pilus assembly motors, they lay claim to more than they should. While this is not a review article and one does not expect a fulminant review of all the T4P motor protein structures and their conclusions, a few more acknowledgements of what has come before and has added to our understanding would be appropriate, in addition to the two cited McCallum papers. The C2 packing, and three states of the nucleotide binding pocket have been reported before. Some potential sources are:

Solanki, Kapoor and Thakur; FEBS J 2018 doi: 10.1111/febs.14619

Nayak, Singh, Zhao, Samsó, Donnenberg mBio 2022 doi: 10.1128/mbio.02270-22

Collins, Karuppiah, Siebert, Dajani, Thistlethwaite, Jeremy Derrick Sci Rep 2018 doi: 10.1038/s41598-018-32218-3

Satyshur et al., Structure 2007 DOI 10.1016/j.str.2007.01.018

Misic, Satyshur, and Forest, JMB 2010 doi:10.1016/j.jmb.2010.05.066

We thank the reviewer for pointing this out and recognize the value of acknowledging the breadth of prior structural studies in the field. We have now included a summary of the conformations and active site pocket occupancies of PilB and PilT in paragraph three of the introduction (lines 56-60) with references to most of the studies cited above.

4. The Interchain contact that mediates nucleotide binding eg. the Arginine Fingers is nicely described. But see Satyshur et al. 2007 description of conserved arginines in adjacent subunit and mutation of conserved LRAAL*R*E (R347 equivalent) that prevents retraction when installed in *P. aeruginosa* PilT at equivalent R194 (re Suppl. Fig 10b). This 2007 paper already describes the interchain contact, shows low resolution electron density of *A. aeolicus* PilT Arg207 with modeled PO4 (Figure 6) and points out, “A cluster of acidic residues is also found near bound nucleotide at the subunit interface in structures of the VirB11 ATPase HP0525, in which it was proposed that ATP prevents the electrostatic clash among the arginines, thus allowing domain closure (Yeo et al., 2000). Interestingly, only Arg207 is invariant; the specific position of neighboring arginines is different in PilT versus the VirB11 ATPases (Figure 1). To our knowledge, this is the first time the arginine finger role has been suggested in PilT or any other member of the type II/type IV secretion ATPase family. Although our results do not permit us to draw detailed conclusions about the role of Arg207 in subunit interactions, they do provide testable hypotheses regarding the roles of the arginine wire in pilus retraction.” This paper also previews Yen *et al.* by suggesting, “that an arginine wire may lead the released Pi product out of the active site”.

We thank the reviewer for bringing our attention to this important publication, which certainly precedes our study and adds value to the role of the arginine fingers mediating nucleotide

interactions. We have expanded on this insight from PilT in paragraph four of the Discussion section (lines 460-470) to provide a more comprehensive analysis of the role that arginine fingers play in PilT/VirB11-like ATPases.

5. AlphaFold multimer modeling including the PilA pilin subunits is bold. But, the motor protein came back as a C6 hexamer and the pilin subunits did not match the cryoEM structure of Sonani *et al.* (Figure 6A, left side). For this reason, the authors very reasonably created a composite model based on experimentally-determined structures. Thus, Panel A, left side should be left out altogether.

We agree and have removed this prediction from Figure 6A, left side. However, since the platform complex from this prediction was used to generate the composite model, we decided to keep it in Supplementary Figure 13 for reference.

6. The AlphaFold structures which are the most reliable (all blue in Suppl Figure 13) presumably are so because there are deposited crystal structures included in the training set. Ideally these should be cited and/or this point should be explicit when comparing the AlphaFold models.

Although AlphaFold undoubtedly queried the PDB for structural orthologs, it currently operates as a "black box" and thus it remains uncertain which structures were utilized for the prediction. There are many such deposited structures in the PDB, including AAA⁺ ATPases which share similar structural folds to PilT/VirB11-like ATPases in their C-terminal ATPase domains, making it challenging to reference all possible structures that could have been utilized for the prediction, and disingenuous to select only a handful that could have been used. We have added a sentence in the last results section (lines 404-407) acknowledging that there are many ATPase structures in the PDB that likely contributed to the better modelling outcomes of the ATPase predictions relative to the platform proteins.

7. The model for retraction is proposed to be a transition from Compact hexamer to Expanded hexamer and back to Compact hexamer, as seen in Figure 3 (which is busy; can the many vector arrow structures on the top line of Panel B be left out?) and Figure 8 (which is fantastic). During the first transition, two subunits which are "Compact-Apo" bind to ATP to form Exp-ATP while two subunits which are bound to ATP (Com-ATP) hydrolyze to become Exp-ADP. Then in the transition back to Compact, the original two ADP sites release nucleotide to become Com-Apo.

We think there is still value in retaining the vector arrows in Figure 3 but do agree that it is a busy figure overall, thus, we have moved the arrow diagrams to the supplementary information (now Supplementary Figure 11). In addition, we have altered our manuscript title to "Coordinated conformational changes in the motor ATPase CpaF facilitate a rotary mechanism of Tad pilus assembly" to more accurately reflect the Tad pilus assembly mechanism proposed in this study.

"Closed, compact and expanded" hexamers are seen, with closed only observed in the absence of nucleotide. The biological role if any of the closed state was not clear. Is the "Closed" hexameric CpaF not part of the normal in-vivo rotary motor trajectory for pilus assembly, serving as a paused state in the absence of ATP or perhaps during a regulated phase when assembly is halted by interaction with another protein?

We have added a paragraph to the Discussion (lines 522-534) addressing the potential biological role of the closed state. Briefly, we propose that the closed state does not actively participate in processive catalysis as it was only observed when prepared in the absence of exogenous nucleotide, and no active site nucleotide densities were identified for this conformation (Figure 1C). The spatial position of the three-helix bundles (3HBs) also likely precludes binding with the platform proteins. Thus, we hypothesize that the closed structure could represent a pre-assembly state when CpaF is trafficking to the poles. The sharpened map showed a fragmented density that could correspond to low occupancy ATP (Supplementary Figure 8A), suggesting that binding of an additional pair of ATP in the adjacent packing unit could induce a closed-to-compact transition. The compact state can then participate in Tad pilus assembly, as per our model (Figure 8). To help illustrate these points, we have added Supplementary Figure 16, which depicts how the ATP binding and hydrolysis mechanism proposed in Figure 3 can transition the closed structure to the compact state.

8. The idea of the supplemental movies is great, but only the full hexamer view turns out to be helpful. The others need less blurry monomer drawings; perhaps less wide cartoon elements.

The monomer movies have been remade so that the domain movements are in context of the full hexamer in a higher resolution video format.

9. Because alpha 5, loop 8 and extended pore loop are colored light blue in Figure 6B but not shown in Figure 1A, I have a hard time understanding if these elements are within the 3HB itself or within the NTD. Please clarify, perhaps by coloring in Figure 1A, or by listing the amino acid ranges of these secondary structure elements in the text or the legend to Figure 6.

We have included the amino acid ranges for alpha 5, loop 8, and the extended pore loop in the main text (lines 360-361) as well as the legend of Figure 6B (line 1024).

Minor comments:

10. Reference 18, Sonani et al., some names are misspelled.

We have now corrected the citation in the reference section.

Reviewer #2 (Remarks to the Author):

In this manuscript, Yen and co-workers report a structural and functional characterization of the Tad pilus ATPase CpaF, in the bacterium *Caulobacter crescentus*. Strikingly, they have obtained three distinct conformations for the CpaF hexamer, which correlates with different nucleotide states, and allows the authors to propose a complete molecular mechanism for CpaF-mediated filament assembly, supported by very elegant structure-function analyses.

This study forms a significant advance in our understanding of the Tad pilus, a bacterial surface appendage of increasing interest due to its wide-spread prevalence in bacteria, and indeed of the related T4P and archaellum. It is well suited for publication in Nature Communications.

Of course, the “elephant in the room” here is the study from Hohl *et al.*, published a few months ago, and which also reports the structure of CpaF. The authors largely ignore this study, and only mention it in one paragraph, in the discussion section. While we sympathize with the authors’ attempt to conserve the novelty of their work, considering that Hohl *et al.* was published > 4 months ago, ignoring it rather detracts from the message of this current work, and indeed the study reported here forms significant advance from the data reported in Hohl *et al.* Therefore, our main recommendation would be to re-write several sections of the manuscript to take this into account.

We thank the reviewer for their positive comments and support for publication in Nature Communications. We have now re-written several sections as per the reviewer’s recommendations below.

Notably, we recommend the following changes:

1. The introduction should mention the results from Hohl *et al.*, and highlight the outstanding questions that resulted from it (most strikingly, the single conformation state of the CpaF hexamer).

We have now included the main findings from Hohl *et al.* in paragraph five of the introduction section (lines 77-82), which will more clearly indicate how our study builds upon theirs.

2. Supplementary figures 1C and D, as well as 8C, are critical to understand the conformation distribution, and the role of related nucleotide states. These should be part of a main figure.

We agree that Supplementary Figure 1C provides useful quantitative comparisons of the particle distribution and have now moved it to the main figures as Figure 1C.

Considering Supplementary Figure 1D, we already show the nucleotide densities and their associated ligands in Figure 1 for the compact and expanded hexamers, which are used for subsequent structural analyses. Supplementary Figure 1D conveys the same information, with the addition of the expanded hexamer from the under-saturated ATP/ADP dataset that we do not include in our analyses. Given this, and the amount of valuable main figure space it would occupy, we believe this panel should remain as part of the supplementary.

Supplementary Figure 8C is similar to Figure 2, except the former includes 1) the closed structure, which is not as important given our view that it does not participate in catalysis, and 2) the full hexamer structure including the 3HBs. As the focus of Figure 2 is on the structural analysis of the C-terminal ATPase domains and their active site pockets, the 3HB domains have been removed from the models in this figure for clarity. As there may be readers who wish to see how these conformational changes are reflected in the spatial orientations of the 3HB domains, we included Supplementary Figure 8C. Given these points, and the amount of space in the main figures that this panel would occupy, we believe it should also remain as part of the supplementary information.

3. Similarly, the structure-function analyses (figure 5) are critical, and the corresponding data (Figure S11A/B, S12B/C) should be included in the main figures.

We believe that moving all these pieces of data to the main figures will unnecessarily crowd Figure 5, and that providing representative microscopy images of all the strains analysed in the main figures is not necessary for comprehension of the experimental outcomes. We have moved Supplementary Figure 11A-B to Figure 5 because we agree that there is value in showing the data for

the chromosomal IDR truncation mutants of CpaF, which are expressed at their native levels and in their native regulatory context. Readers who are curious about the microscopy images for the overexpressed IDR truncation mutants can still find them in Supplementary Figure 12.

4. In contrast, the phylogenetic analyses (Figure 4 and 7) do not really contribute much to this work, and rather distract from its main message. These should be relegated to the supplementary material, and the corresponding results section should be shortened.

We believe that the phylogenetic analysis in Figure 4 adds value to our manuscript, a point that was raised by other reviewers. Hohl *et al*¹. suggested that the intrinsically disordered region (IDR) may allosterically modulate CpaF bidirectionality, a hypothesis that our data disagrees with. The phylogenetic analysis (Figure 4) suggests that the majority of CpaF orthologs do not have an IDR. Therefore, it is unlikely that the IDR contributes to the core activity of CpaF. Our pilus labeling and blocking experiments in Figure 5 further support our hypothesis.

Figure 7 contributes to our hypothesis, and that of others that study type IV pili (T4P), that the experimentally observed conformational changes in the ATPase are conveyed to the platforms, which in turn undergo as-yet undetermined conformational changes that form the basis of how the pilus filament is assembled and disassembled from the major pilin subunits. A key detail of this process is likely to be how the ATPase contacts the platforms. Given that members of the type IV filament (TFF) superfamily are evolutionarily conserved, we sought to address this question by using newer modelling methodologies (AF3) to predict 1) how CpaF contacts the Tad platform proteins and 2) if these points of contact are similar in other TFF superfamily members. The predictions in Figure 7 show that the points of contact between the ATPase and platform proteins are conserved, suggesting that insights from our study (for example, Figure 8) may be applicable to other TFF systems. Figure 7 also highlights the remarkable structural and architectural conservation of the TFF motor complexes generally, supporting prior analyses which have proposed an evolutionary relationship between members of this family using phylogenetic approaches². We believe these findings are of general interest to the community that studies TFF systems, including T4P, T2SS, archaeal pili, and the archaellum, and therefore inclusion of this figure in the main text is justified.

Other major comments that need to be addressed:

5. The authors go to great lengths to resolve the conformation of the 3HB domain in their maps, and indeed its conformation varies very significantly between the different conformations. Yet it is not involved in the interaction with the nucleotide, and indeed the modelling suggests that its role may be to interact with CpaG and CpaH (lines 364-370). A more in-depth analysis of the different positions observed for this domain, how it may be affected by the conformation of the hexamer, and how it may affect interaction with the assembly machinery and/or the pilin protein, would be beneficial.

We performed a more in-depth analysis of the spatial position of the 3HBs in each of the three structures by measuring the C α -C α distances of D116 (the same residue used for the measurements in Figure 3D) between symmetrically positioned 3HBs as well as between adjacent 3HBs (now Supplementary Fig. 1B). The former provides a quantitative definition of the “closed”, “compact”,

and “expanded” conformations, while the latter highlights which 3HBs undergo the largest change during the nucleotide-driven transition from one conformation to the next (lines 136-139). These distances are described in the context of the CpaF catalytic mechanism by indicating how the rotation of the 3HB and NTD as one rigid body during ATP hydrolysis increases the 3HB distances between adjacent chains as well as across chains (lines 255-256). In the results section detailing the AlphaFold3 prediction of the Tad motor subcomplex, we expand on the predicted interactions between the 3HB and CpaGH to suggest that the motions experienced by the 3HB (Supplementary Fig. 1B) during catalysis could influence the conformations of the N-terminal $\alpha 1$ of CpaH, which appears to gate the 22 Å wide opening between CpaGH heterodimers (lines 380-382, 391-394). We additionally state that the compact and expanded hexamer pore can accommodate the platform complex, though some conformational rearrangement is necessary in the compact state (lines 371-374).

6. The ATPase activity assay reported in Figure 5 is important as it establishes that the CpaF hexamer is functional in isolation, validating the different conformations observed. However, this experiment requires a negative control, with a mutant lacking catalytic activity. Ideally, a positive control, with another well-characterized ATPase, could also be used.

We performed the ATPase activity (Panel A below) and thermal shift (Panel B below) assays with the addition of a catalytically inactive mutant in the Walker A motif, CpaF^{K287A} as a negative control and the type IVa pilus (T4aP) retraction ATPase PilT from *Geobacter metallireducens* with

A

B

known ATPase activity³ as a positive control. The data for CpaF^{K287A} is now included in the revised Figure 5B-C in the main text. However, due to the drastic differences between the relative activities and thermal stabilities of CpaF and PilT, we have decided that it is not an appropriate

positive control and did not include the PilT data in the manuscript.

Panels A & B (above). **A.** *In vitro* enzyme-coupled ATPase assay of purified CpaF IDR truncation constructs and *G. metallireducens* PilT. Mean ATPase activity was determined from three biological replicates and activity depicted as the change in nM of NADH per min per mg of protein. **B.** *In vitro* thermal shift assay of purified CpaF IDR truncation constructs and *G. metallireducens* PilT. Melting temperature was determined from four biological replicates of a gradient denaturation

using SYPRO orange from 10-95°C with 0.5°C incremental increases per 30 s. For panels A and B, each data point represents one biological replicate with three technical replicates. Statistics were determined using Dunnett's multiple comparisons test. *** $p < 0.001$. **** $p < 0.0001$. ns, not significant. Error bars show standard error of the mean (SEM).

Additional minor comments:

7. Line 27: As powerful as AF3 is, we don't think it is accurate to say that it "demonstrates" something, but rather suggest/indicates.

We agree and have altered the wording to "suggest".

8. Line 35: we find the term TFF rather confusing, and although it has been employed in Denise et al (PloS Biol, 2019),

We think the reviewer did not complete this thought and cannot fully address this comment. We are following the existing nomenclature first introduced by Berry & Pelicic⁴ and subsequently adopted by Denise *et al.*² and feel that it is unnecessary to modify this terminology. The CpaF study by Hohl *et al* also utilizes this nomenclature¹.

9. Line 101: What is the rationale for using ADP and ATP at under-saturating condition? Obviously it produced a very interesting result, but is this a common/previously used strategy to characterize AAA+ ATPases? If so, can the authors provide references?

Two crystal structures of the T4aP ATPase PilB from *G. metallireducens*⁵ and *Thermus thermophilus*⁶ were determined independently by two different groups. Their differing sample preparation methods, one saturated with ATP γ S and the other under-saturated with AMP-PNP, resulted in two structures with different nucleotide occupancies. In the PilB structure saturated with ATP γ S, all six monomers were occupied by the ligand. Under-saturating AMP-PNP led to a structure with four AMP-PNPs and two ADPs bound in the active sites. The authors of these two studies consequently proposed different catalytic mechanisms. When we similarly under-saturated and saturated the CpaF active sites, we observed an additional compact state bound to four nucleotides, which enabled us to propose our rotary mechanism of CpaF catalysis.

10. Lines 338-339, and 344-346: AF3 cannot model helical assemblies, therefore including PilA here was not going to work. This should not be included.

We have removed the AlphaFold3 modeling with PilA (Figure 6A, left side). However, since the platform complex from this prediction was used to generate the composite model, we decided to keep it in Supplementary Figure 13 as a reference.

11. Lines 502-505: An additional figure, perhaps as part of Figure 8, comparing the mechanisms of CpaF to that of PilB and PilT, would help illustrate this.

We have added a new figure Supplementary Figure 15 to illustrate the differing rotary mechanisms employed by PilB and PilT. We have also included the mechanism proposed by Hohl *et al*¹ to this figure to highlight the difference between the proposed CpaF catalytic mechanisms.

12. Lines 99, 106, 142: The authors use unsharpened maps to analyse nucleotide density. Considering the authors' expertise, we have no doubt that it is the correct thing to do – however this is not standard procedure these days. We encourage the authors to add a sentence or two to explain the rationale here, with reference if appropriate.

We chose to show the nucleotide density from the unsharpened maps to prevent biasing the experimental reconstructions from global over-sharpening that is automatically applied in cryoSPARC. Given our maps refined to sufficiently high resolution, we were able to differentiate the nucleotides from the densities. However, to ensure we follow standard practice in the field, we have opted to show the locally sharpened nucleotide densities in Figure 1E and Supplementary Figure 1D.

Reviewer #3 (Remarks to the Author):

Reviewer #4 (Remarks to the Author):

This review has been written with the help of a graduate student for training purpose with approval by the editor.

The manuscript by Yen *et al.* analyzes the structure of the pilus motor complex CpaF in *C. crescentus* by cryoEM and how nucleotide exchange yields conformational changes that drive function. CpaF is a member of the single motor type IV filament superfamily (TFF), which facilitates extension and retraction of type IV pilus (TFP) filaments. This is in contrast with other members of this family that have dedicated extension and retraction ATPases. The process of how this single motor facilitates both directions is unknown, and the current manuscript presents important insights into this question. The manuscript has done an excellent job thoroughly explaining CpaF structure and mechanism of rotation in a way that is easy to follow and well written. The diverse combination of experimental data, structural analysis, dynamics, and modeling implemented by the authors is key for understanding how these complexes function in-vivo and interact with the rest of the TFP machines.

We are not experts in protein structure or cryoEM and leave judgement of the details to other reviewers, but the results and conclusions appear technically sound to us. We are experts for functional in-vivo assays of TFP dynamics as shown Fig. 5. These data are a strength of this manuscript and aid to the structural data (structure papers are often lacking experimental connections to in-vivo). As the authors point out, their findings and proposed model for how this motor class drives TFP extension/retraction has broad implications for the large field of TFF systems.

We would like to point out that the current manuscript adds much in rigor, depth, and quality compared to reference 31, supported by more detailed experiments that yield better insights and understanding of how the CpaF system might work.

We only have several minor comments and recommend publication after these have been addressed.

We thank the reviewers for their positive comments on our manuscript and support for publication.

Minor comments:

1. The manuscript refers to “PilT-like structure” several times. It is unclear why not compare to a PilB-like structure?

This is a fair question. The nomenclature “PilT-like structure” has been previously referenced in other manuscripts to encompass the PilT/VirB11-like ATPase family that is structurally distinct from the AAA+ ATPase family^{3,5,7,8}. We now refer to these ATPases as “PilT/VirB11-like ATPases” throughout the main text and abstract.

2. cryoEM is used as abbreviation for “electron cryo-microscopy”, which seems to be a mix-up in order of words in the abstract and introduction

Another great observation. Our understanding is that the proper nomenclature for cryo-EM remains up for debate in the field⁹. However, to avoid confusion for readers, we have changed the abbreviation ‘cryo-EM’ to mean ‘cryo-electron microscopy’.

3. Some language, especially in the beginning of the results, uses jargon that is not easy to follow for non cryoEM or structure experts. Please explain some terminology, for example C₂ symmetry, what are particles (especially, are these monomers or oligomers), the reference of chains a and d (line 95-96), what does over-refined mean (line 123).

We have attempted to explain these terms here and in the main text:

- C₂ symmetry refers to two-fold rotational symmetry, which means a 180° rotation of the CpaF hexamer about the central axis (i.e. through the ‘hole’ in the center of the hexamer) will produce the identical structure. We have now defined C₂ as “two-fold rotational symmetry” when first mentioned in the introduction section (line 57).
- CpaF purifies as a hexamer, thus whenever we refer to particles from a cryo-EM perspective, we are always referring to a hexamer. We have clarified this at the first instance when “particles” is mentioned: “Thirty-six percent of the hexameric particles...” (lines 107-108).
- Chains refer to the six monomers (a-f) that comprise the CpaF hexamer. We use the distance between the three-helix bundles (3HBs) of chains a and d (Supplementary Figure 1B) as a means of differentiating the three CpaF conformations observed in our structures. To avoid confusion, we now say: “In the sample without exogenous nucleotides, we resolved two distinct C₂ symmetric structures, which we termed “closed” and “compact” based on the distance between symmetrically positioned residue D116 on the 3HBs, to overall resolutions of 3.4 and 3.2 Å, respectively” (lines 104-107).
- Over-refinement refers to excessive integration of noise into the cryo-EM map during particle alignment, generating artificial features that do not represent the true biological molecule. We believe over-refinement of the 3HB densities originates from their inherent

flexibility, which makes particle alignment challenging. We have added a short description in the main text “emergence of artificial features in the map” (lines 132-133).

4. Line 178 – “walker A G284” – this is the only thing that is not labeled on the graph it would be good to add it in

We have now labeled this residue in Figure 2 and Supplementary Figure 9.

5. Lines 233-252: This description was challenging to follow. Please consider rephrasing.

We have rephrased the descriptions for ATP hydrolysis (now lines 245-256) and ADP release (now lines 257-266). The corresponding Figure 3E-H was also slightly modified to incorporate the angular values of domain rotation and distance of translation.

6. Lines 253-260: If CpaF always rotates in the same direction, how can it facilitate extension and retraction?

This question was similarly brought up by other reviewers and unfortunately our structural and functional analyses of CpaF alone are unable to address this. The conformational changes that we have described in the section “The CpaF hexamer expands and contracts during the catalytic cycle” and subsequent proposal of CpaF operating under a clockwise, rotary mechanism likely apply to Tad pilus assembly given that a similar mechanism was proposed for the type IVa pilus assembly ATPase PilB⁵. We have clarified this in lines 271-274 and further expanded on this in the discussion (lines 548-560). The mechanism of how CpaF could facilitate rotation of the platform proteins in the opposite direction remains elusive.

7. Figure 3D: the coloring of the in-figure legend is ambiguous. Please either change colors or add corresponding bars with/without patterns in front of each item in the legend.

Figure 3D now includes bars with/without patterns in front of each residue in the legend.

8. Fig 7A: The docking of N/C terminal domains seems to be different between the different systems. Can the authors elaborate?

All the motor-platform predictions were performed using AlphaFold3¹⁰. Of the six predictions performed, the PilT-PilC prediction was the sole candidate where the ATPase was oriented “upside down” (i.e. with the C-terminal ATPase domain facing the platforms). We make note of this observation in lines 407-411 and further denote that the previously characterized AIRNLIRE motif in the C-terminal ATPase domain of PilT likely resulted in the different prediction outcomes¹¹. We did not perform any manual docking in Figure 7.

9. Since one of the main premises of the paper is to understand how a single motor can drive extension and retraction, can the authors elaborate more on this point? Could it be that CpaG and CpaH each mediate one direction and are exchanged between extension/retraction?

We are moving away from this premise as we present insufficient evidence in our manuscript to fully address this question. We were unable to pinpoint any regions of CpaF that differentiated Tad pilus assembly from disassembly, and indeed a previous random UV mutagenesis screen probing for retraction-deficient mutants in *cpaF* failed to do the same¹². This suggests that CpaF likely is

not the candidate that mediates switching between Tad pilus extension and retraction, despite its requirement for both functions. We believe other Tad components that are predicted to interact with CpaF, including CpaG and CpaH, likely fill this role and have provided a short discussion in lines 548-560. AlphaFold3 predicts that three copies each of CpaG and CpaH form a trimeric platform architecture that can accommodate three pilin subunits at the base (Figure 6E). It is possible that these pilin subunits differentially interact with CpaG and CpaH during Tad pilus assembly and disassembly (i.e. CpaG = assembly and CpaH = disassembly, or vice versa).

Reviewer #5 (Remarks to the Author):

REFERENCES

1. Hohl, M., Banks, E. J., Manley, M. P., Le, T. B. K. & Low, H. H. Bidirectional pilus processing in the Tad pilus system motor CpaF. *Nat Commun* **15**, 6635 (2024).
2. Denise, R., Abby, S. S. & Rocha, E. P. C. Diversification of the type IV filament superfamily into machines for adhesion, protein secretion, DNA uptake, and motility. *PLoS Biol* **17**, e3000390 (2019).
3. McCallum, M. *et al.* Multiple conformations facilitate PilT function in the type IV pilus. *Nat Commun* **10**, (2019).
4. Berry, J.-L. & Pelicic, V. Exceptionally widespread nanomachines composed of type IV pilins: the prokaryotic Swiss Army knives. *FEMS Microbiol Rev* **39**, 134–154 (2015).
5. McCallum, M., Tammam, S., Khan, A., Burrows, L. L. & Howell, P. L. The molecular mechanism of the type IVa pilus motors. *Nat Commun* **8**, (2017).
6. Mancl, J. M., Black, W. P., Robinson, H., Yang, Z. & Schubot, F. D. Crystal Structure of a Type IV Pilus Assembly ATPase: Insights into the Molecular Mechanism of PilB from *Thermus thermophilus*. *Structure* **24**, 1886–1897 (2016).
7. Reindl, S. *et al.* Insights into FlaI Functions in Archaeal Motor Assembly and Motility from Structures, Conformations, and Genetics. *Mol Cell* **49**, 1069–1082 (2013).
8. Iyer, L. M., Makarova, K. S., Koonin, E. V & Aravind, L. Comparative genomics of the FtsK-HerA superfamily of pumping ATPases: implications for the origins of chromosome segregation, cell division and viral capsid packaging. *Nucleic Acids Res* **32**, 5260–5279 (2004).
9. Henderson, R. & Hasnain, S. ‘Cryo-EM’: electron cryomicroscopy, cryo electron microscopy or something else? *IUCrJ* **10**, 519–520 (2023).
10. Abramson, J. *et al.* Accurate structure prediction of biomolecular interactions with AlphaFold 3. *Nature* **630**, 493–500 (2024).
11. Aukema, K. G., Kron, E. M., Herdendorf, T. J. & Forest, K. T. Functional dissection of a conserved motif within the pilus retraction protein PilT. *J Bacteriol* **187**, 611–618 (2005).
12. Ellison, C. K. *et al.* A bifunctional ATPase drives tad pilus extension and retraction. *Sci Adv* **5**, (2019).

REVIEWERS' COMMENTS

Reviewer #1 (Remarks to the Author):

The authors have done a thorough job responding appropriately to all of the reviewers' comments.

Reviewer #2 (Remarks to the Author):

The authors have extensively addressed the comments from all three reviewers, and as a consequence the revised manuscript is a lot stronger. We are now delighted to recommend its publication in Nature Communications.

If possible, we would recommend the following minor adjustment:

- We agree with the authors' suggestion not to include the PilT control in the main figure, as the K287A mutant is a very strong negative control, which very elegantly support their structure. However, the fact that CpaF is a lot more active than PilT, in isolation, is a very interesting observation, which could be related to the bi-directional role of CpaF? If there is enough room, I would recommend that the authors add this as a panel in the supplementary material, and include a sentence mentioning this.

PilT was only meant to serve as a positive control for the ATPase assay and thus we never intended to make direct comparisons of its activity to CpaF. It is also important to note that the assay conditions may not be optimal for *G. metallireducens* PilT, leading to either an under- or overestimation of its activity. Finally, given that we were unable to differentiate Tad pilus assembly and disassembly from the CpaF structures, it remains unclear whether the ATPase activity observed in the assay would power polymerization, depolymerization, or both. Thus, we believe directly comparing their activities may be misleading.

Reviewer #3 (Remarks to the Author):

Reviewer #4 (Remarks to the Author):

The authors have addressed all of our concerns.

Reviewer #5 (Remarks to the Author):
